# Single-cell transcriptomes identify patient-tailored therapies for selective co-inhibition of cancer clones

Aleksandr Ianevski [1,10], Kristen Nader[1,10], Kyriaki Driva[2], Wojciech Senkowski [2], Daria Bulanova[1,2], Lidia Moyano-Galceran [2], Tanja Ruokoranta [1,3], Heikki Kuusanmäki[1,2,4], Nemo Ikonen [1], Philipp Sergeev[1], Markus Vähä-Koskela[1], Anil K. Giri [1,4], Anna Vähärautio [4,5,6], Mika Kontro[1,3,4], Kimmo Porkka [3,6], Esa Pitkänen [1,6,7], Caroline A. Heckman [1,6], Krister Wennerberg [2] & Tero Aittokallio [1,6,8,9] ✉

Intratumoral cellular heterogeneity necessitates multi-targeting therapies for improved clinical benefits in advanced malignancies. However, systematic identification of patient-specific treatments that selectively co-inhibit cancerous cell populations poses a combinatorial challenge, since the number of possible drug-dose combinations vastly exceeds what could be tested in patient cells. Here, we describe a machine learning approach, scTherapy, which leverages single-cell transcriptomic profiles to prioritize multi-targeting treatment options for individual patients with hematological cancers or solid tumors. Patient-specific treatments reveal a wide spectrum of co-inhibitors of multiple biological pathways predicted for primary cells from heterogenous cohorts of patients with acute myeloid leukemia and high-grade serous ovarian carcinoma, each with unique resistance patterns and synergy mechanisms. Experimental validations confirm that 96% of the multi-targeting treatments exhibit selective efficacy or synergy, and 83% demonstrate low toxicity to normal cells, highlighting their potential for therapeutic efficacy and safety. In a pan-cancer analysis across five cancer types, 25% of the predicted treatments are shared among the patients of the same tumor type, while 19% of the treatments are patient-specific. Our approach provides a widely-applicable strategy to identify personalized treatment regimens that selectively co-inhibit malignant cells and avoid inhibition of non-cancerous cells, thereby increasing their likelihood for clinical success.

High intratumoral cellular heterogeneity and clonal evolution of cancer cell populations are major drivers of therapy resistance both in hematological malignancies and solid tumors[1–5]. In acute myeloid leukemia (AML), several single-cell genomic analyses have mapped the clonal evolutionary processes of disease progression and therapy resistance at the cell subpopulation level, as well as deciphered cellular hierarchy and reprogramming among the leukemic cell subpopulations involved in chemoresistance, relapse and clinical outcomes[6–9]. Similarly in solid tumors, intratumoral and interpatient heterogeneity are significant medical challenges both for disease diagnosis and treatment optimization. The highly heterogeneous tumor ecosystem contains not only malignant cells but also other cell types, such as

endothelial cells, stromal fibroblasts, and a variety of immune cells that control tumor growth and invasion. Notable studies in melanoma, ovarian and colorectal cancers have demonstrated that specific characteristics of the tumor immune microenvironment (TME) can, to some extent, predict a patient's clinical outcome[10]. For instance, clonal analysis and longitudinal sampling of patients with high-grade serous ovarian carcinoma (HGSC) revealed evolutionary trajectories, with distinct genomic and morphological features across patients that associate with therapy responses[11]. Moreover, the interaction between immune and non-immune cells within the TME can influence the effectiveness of immune responses, leading to varied treatment outcomes among patients.

Single-cell analyzes in cancer research utilize a wide array of advanced techniques aimed at understanding the heterogeneity within tumors at the individual cell level. These methodologies provide insights into the genomic, transcriptomic, and epigenomic variations among cancerous and healthy cells, offering a more comprehensive understanding of cancer biology and progression. While multiple single-cell technologies have been developed, scRNA-seq is currently the most popular and matured technology, and it allows researchers to analyze the transcriptome of individual cells, identifying gene expression patterns and heterogeneity across cell populations. Advances in scRNA-seq technology have improved its sensitivity and accuracy, enabling the characterization of even rare cell populations in both tumor and TME. Single-cell analyses are increasingly being applied in clinical settings for precision oncology. Profiling individual cells allows for the identification of specific biomarkers and the development of personalized treatment strategies tailored to the unique characteristics of a patient's cancer. Single-cell technologies have the potential to open up new applications in cancer research; however, translational precision medicine strategies that use single-cell data are still rare[12]. Current challenges in clinical applications include the robustness, scalability, and cost-effectiveness of single-cell assays when profiling complex patient samples.

Tumor-specific drug combinations are often required to provide clinical benefits for patients with advanced, relapsed, or refractory malignancies[13,14]. However, there is a medical need for systematic approaches to identify more effective combinatorial therapies, using either multi-targeting inhibitors or their combinations, which selectively co-inhibit multiple signaling pathways that drive the disease- or resistance in heterogeneous patient and cell populations. Despite the wealth of information on cancer evolution and intra-tumoral cellular heterogeneity, we lack approaches that target chemoresistant subpopulations to enhance second-line treatment efficacy in relapsed patients or to avoid resistance to first-line therapies by co-inhibiting multiple cancer cell subpopulations with sufficient high potency and precision. Several computational approaches have been developed that use scRNA-seq data to associate individual cells with disease attributes, such as diagnosis, prognosis, and response to therapy[15–20]; however, none of these methods enable the identification of multi-targeting drugs or drug combinations for genetic clones at a single-cell and individual-patient level. In particular, there is a lack of approaches that consider both the patient and disease heterogeneity when predicting drug sensitivity differences among cell populations, with the aim to design cancer-selective and patient-specific therapeutic options using computationally and experimentally scalable and clinically feasible profiling measurements in scarce patient-derived primary cells. The use of large-scale drug testing data for predictions poses a practical challenge, since systematic ex vivo drug testing in primary patient cells is currently not feasible in many solid tumor types[21].

To address these limitations, we present a machine learning model, scTherapy, which identifies cancer-selective and low-toxic multi-targeting options for each individual cancer patient based on a single scRNA-seq count matrix alone. The selective predictions originate from transcriptomic differences between genetically distinct cancer cell populations (or clones) in individual patient samples when compared to non-cancerous cells from the same patient sample. To enable fast translational applications, we pre-train a gradient boosting model (LightGBM) that learns drug response differences across cell populations by leveraging a massive reference database of large-scale phenotypic profiles (both transcriptomics and viability readouts) measured in cancer cell lines in response to single-drug perturbations. When applied to a patient sample, the model generates a ranked list of the most effective multi-targeting options (either targeted agents, chemotherapies, or their combinations) that selectively co-inhibit key cancer clones in each individual patient sample. To guide translational applications, we further remove low-confidence predictions and non-tolerated doses among the dose-specific drug response predictions, hence ensuring that only the most relevant predictions will be suggested for treatment optimization. The scTherapy predictions makes ex vivo drug testing in patient-derived cells more cost-effective by prioritizing the most potent multi-targeting options for further experimental validation in scarce patient cells. Moreover, we expand the combinatorial space of single-cell drug response assays, currently constrained by the excessive time and cost of the assays for translational use.

## Results

To design multi-clone targeting and cancer-selective therapeutic options for each patient, we leveraged 394,303 genome-wide transcriptomic profiles post-treatment with 19,646 single-agent responses, measured in multiple doses in 167 cell lines, available from the LINCS 2020 project[22]. We next matched these transcriptomic response profiles with drug-induced cell viability responses available from PharmacoDB[23], measured in multiple doses in the same 167 cell lines to pre-train a LightGBM that predicts drug response differences across cell populations (Fig. 1, **Online Methods**). The model predicts drug response using fold changes of differentially expressed genes (DEGs) after drug treatment at a particular dose, hence leading to concentration-specific cell inhibition predictions. In the patient applications, we used the pre-trained model to predict multi-targeting options that can selectively co-inhibit multiple cancer subclones, identified from patient-specific scRNA data, and using fold changes of DEGs between normal cells and cancer cell populations as input. In the final step, we combined the top-predicted effective and selective drugs for each clone as a targeted combinatorial therapy for the patient sample. This translational approach enables the systematic tailoring of personalized multi-targeting options by considering both the intratumoral cellular heterogeneity and dose-specific therapeutic and toxic effects of anticancer compounds.

### Experimental validation of the model predictions in AML patient samples ex vivo

We developed the scTherapy model and tested its translational potential first by analyzing single-cell transcriptomic profiles of 12 bone marrow samples from diagnostic and refractory or relapsed AML patients with various driver mutations and treatment regimens (Suppl. Table 1–2), followed by careful experimental validation of the model predictions in the primary cells of the same patient samples. The single-cell transcriptomes revealed highly heterogeneous cell type compositions across the heterogeneous population of patients and cells of both leukemic and normal cell types (Fig. 2a), necessitating personalized treatment predictions. Through processing the scRNA-seq data from each patient separately, and then feeding these into the pre-trained scTherapy model, we generated personalized predictions of combinatorial drug treatments aiming to selectively target two major subclones in each individual patient (see Online "Methods"). In some cases, the treatment response predictions corresponded to the subsequent clinical treatments of the patients after taking the sample (Suppl. Table 2).

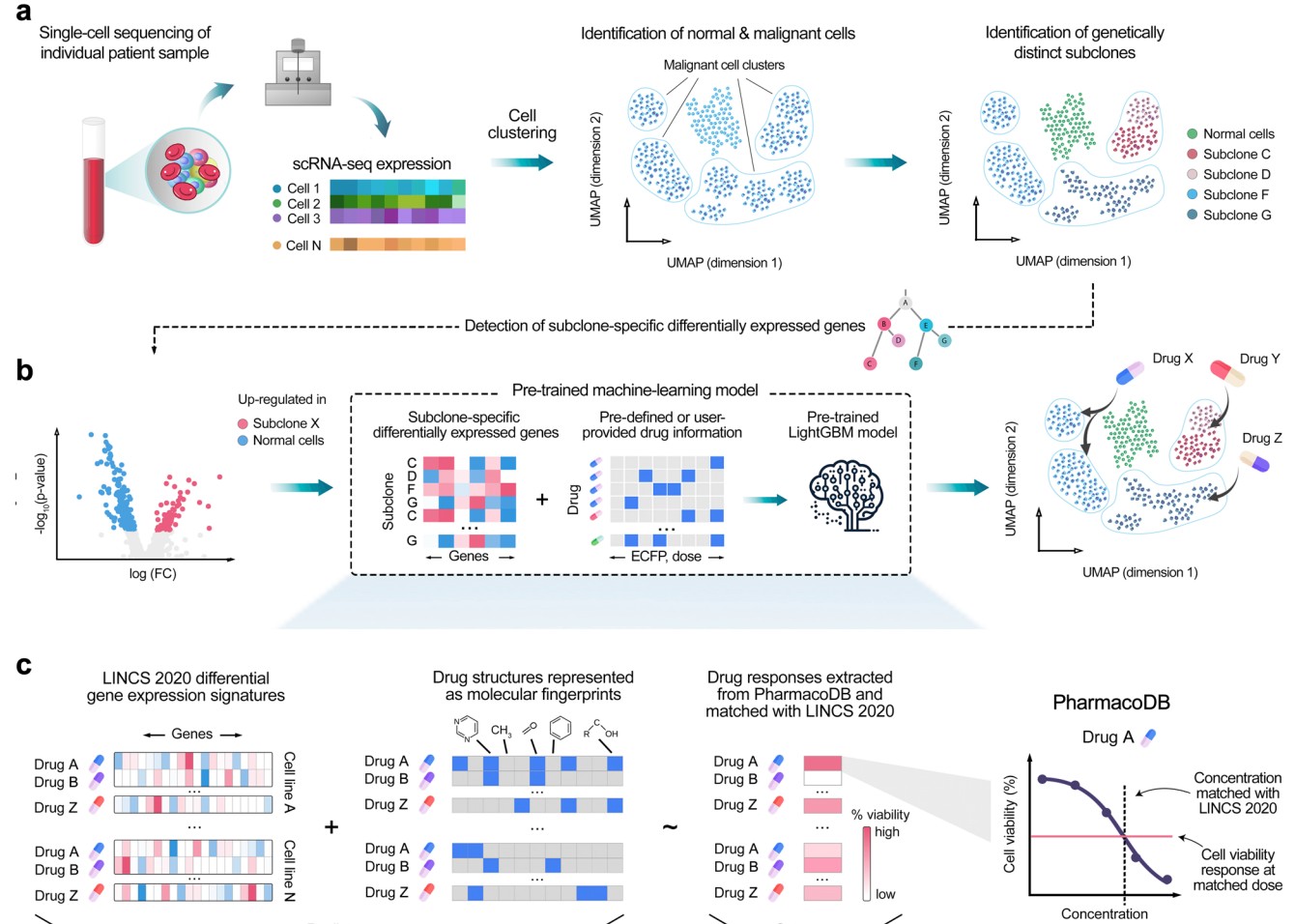

**Fig. 1 | Schematic illustration of the experimental-computational prediction approach.** Identification of clone-specific and cancer-selective compounds is performed in two steps: (**a**) Raw sequencing data from selected tissue are processed and aligned to generate a scRNA-seq expression count matrix. Unsupervised clustering separates malignant and normal cell clusters using an ensemble prediction approach with three analytical tools: ScType, CopyKAT and SCEVAN (see Suppl. Figure 3). InferCNV infers large-scale copy number variations and identifies genetically distinct subclones among the malignant cells. **b** Subsequently, subclone-specific differentially expressed genes are identified through differential expression analysis. The identified genes, along with drug information such as molecular fingerprints and drug doses, serve as inputs for the pre-trained

LightGBM model. Based on the patient-specific inputs, the pre-trained model predicts the most potent compounds and their effective doses for each subclone. **c** To train the LightGBM model, a comprehensive dataset was compiled that integrates transcriptional changes from small-molecule perturbation experiments (LINCS 2020 dataset)[22], with chemical structures represented as ECFP fingerprints and drug-dose response data collected from various studies (PharmacoDB resource)[23]. Concentrations of the LINCS 2020 dataset were matched with dose-response curves from the PharmacoDB, and interpolated cell viability was used as the outcome variable for LightGBM model. scTherapy can propose potential drugs among any of the 3695 unique compounds overlapping with the LINCS 2020 and PharmacoDB resources.

To validate the model predictions, we first used data from single-agent cell viability assays, which confirmed that the model-predicted effective treatments led to significantly better cell inhibition efficacy ex vivo, when compared with the predicted ineffective treatments ($p < 0.0001$, Wilcoxon test; Fig. 2b). Importantly, this improvement was not due to the model selecting higher drug concentrations for the effective-predicted treatments (Suppl. Figure 1a). Most of the treatment predictions were uniquely identified for a single patient, and the few shared treatments between patients, such as navitoclax and AT-7519, showed highly variable responses across the patient samples (Fig. 1b, the colored points). Such treatment response variability is expected in this diverse patient cohort, which spans different disease stages and samples with highly heterogeneous cell type compositions. However, we did not find significant differences either in the number of drug predictions, or in predicted overall effective doses between diagnosis and refractory samples of the same patient (Suppl. Figure 2).

Next, we predicted the most promising two-drug combinations in the four AML patient samples with enough cells for further experimental testing. The patient-specific combinations were designed so

that they would maximally co-inhibit the two major leukemic subclones in each patient sample, while minimally co-inhibiting the patient-specific normal cells (Fig. 2c). Using initially a bulk cell viability assay, we tested the predicted combinations in $4 \times 4$ dose-response matrices (all the patient-specific combination matrices and synergy distributions are provided at https://ianevskialeksandr.github.io/scTherapyCombinations.html). Based on the zero interaction potency (ZIP) score, we confirmed that all the predicted combinations act either synergistically (ZIP > 10), i.e., they jointly inhibit patient cells more than expected based on the single-agent effects ($p < 0.001$, Wilcoxon test), or showed at least additive combination effects (ZIP > 0; Fig. 2d). It has been argued that combination efficacy is more important in practice, while pharmacological synergy is not necessary for achieving improved clinical responses[24].

After confirming the higher than expected combination effects in the bulk viability assays, we further tested a subset of the top-6 patient-specific combinations for the four patient cases using high-throughput flow cytometry assays to quantify the differential inhibition between leukemic and normal cells in each patient sample

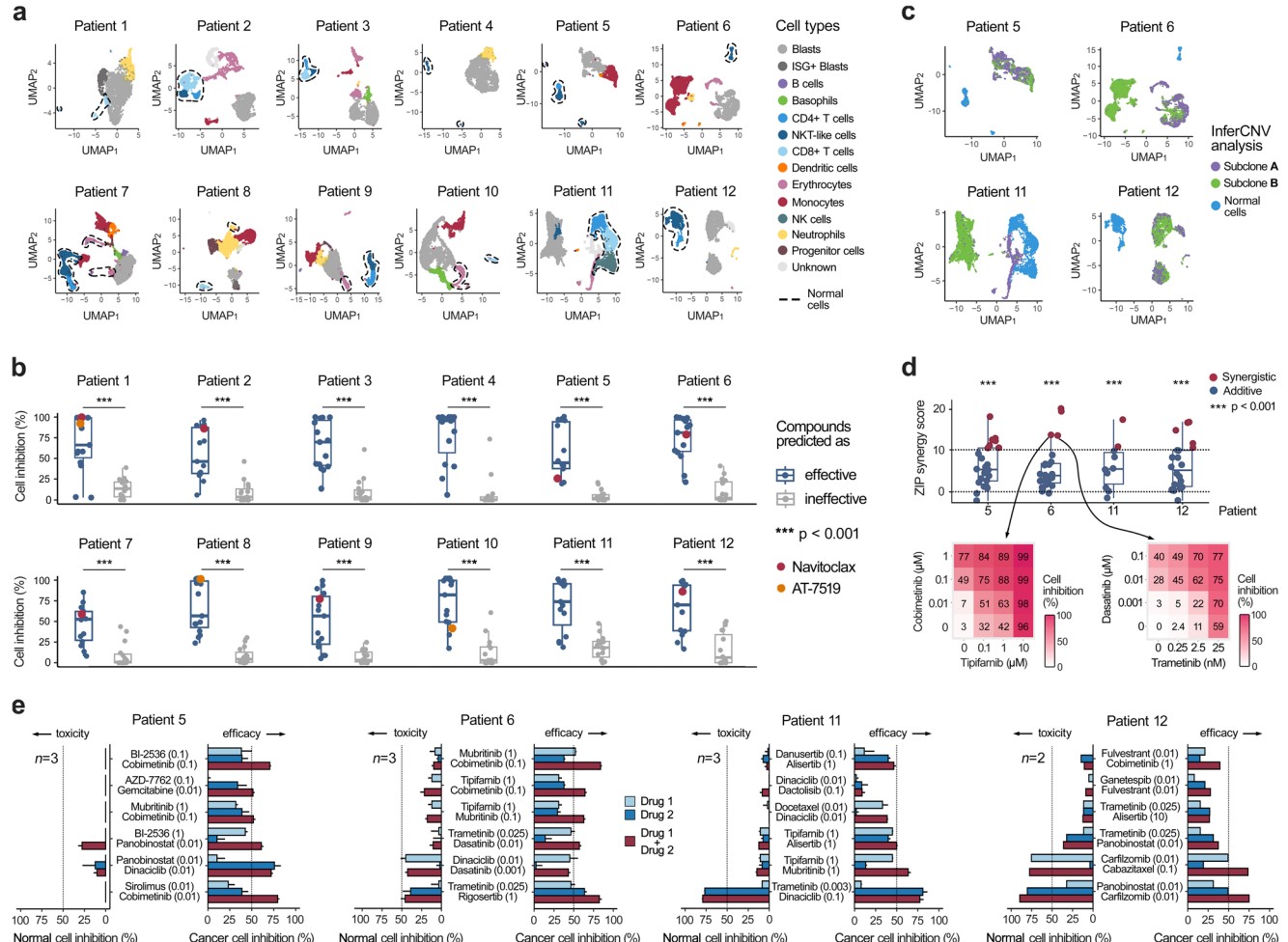

**Fig. 2 | Experimental validation using bulk and cell population drug assays.**
**a** Identification of cell types using scRNA-seq profiles of complex bone marrow samples from 12 AML patients (2140–9340 cells analyzed per patient, Suppl. Table 1). **b** Ex vivo drug sensitivity differences between single-agent treatments predicted by scTherapy to be either effective ($n = 15$) or ineffective ($n = 15$) in whole-well cell viability assays ($p < 0.001$, two-sided Wilcoxon test). The colored points show two example drugs with highly variable responses across the patients. Source data are provided as a Source Data file. **c** Identification of genetically distinct subclones from the 4 AML patient samples with enough cells for further experimental testing (Suppl. Figure 4 shows a detailed overview of genomic variation in these 4 samples). **d** All the model-predicted drug combinations exhibited either synergistic (ZIP > 10) or additive effects (0 < ZIP < 10) in the whole-well combinatorial viability assay. Statistical analysis showed significant effects (p < 0.001, two-sided Wilcoxon test; upper panel), when comparing measured ZIP scores to zero (no effect). The assay involved $n = 24$ combinations for Patients 5 and 6, $n = 11$ combinations for Patient 11, and $n = 20$ for Patient 12, with each patient's results representing technical replicates derived from different drug combinations tested on cells from the respective patients. Two examples of combinations with ZIP = 13.6

and ZIP = 13.5 as tested in multi-dose drug combination assays (lower panel). Interactive plots of the dose-response matrices for all the predicted combinations are provided at https://ianevskialeksandr.github.io/waterfall_plot.html. Scale bars represent percentage inhibition from 0 to 100 (white to red). **e** Further validation of the top-combinations for the 4 patient samples using population-level flow cytometry assays in the same patient-derived cells. Toxic effects (left-hand bars) scored based on co-inhibition of normal cell populations, and therapeutic effects (right-hand bars) based on co-inhibition of malignant cells. The predicted effective doses are indicated in parentheses (μM), and the dotted vertical lines indicate 50% inhibition level. n, the number of replicate screens. Statistically significant differences in combinatorial responses between cancerous and normal cells were observed, with $p < 0.01$, < 0.05, and < 0.05 for patients 5, 6, and 11, as assessed with the two-sided Wilcoxon signed-rank test. For patient 12, given the limited number of biological replicates ($n = 2$), statistical significance cannot be tested. The error bars represent the standard error of the mean (SEM). Box plots show the median (central line), 25th and 75th percentiles (box edges), and the range within 1.5 times the interquartile range from the box (whiskers). Source data are provided as a Source Data file.

ex vivo. Out of the 24 predicted drug combinations, 21 (88%) led to increased co-inhibition of the leukemic cells (Fig. 2e), when compared with the single-agent responses. For each patient case, we identified multiple combinations that led to higher than 50% co-inhibition of the blasts and other leukemic cells, suggested as potential treatment options. Importantly, only 3 of the 24 combinations (13%) showed >50% inhibition of T cells and other non-cancerous lymphoid cells, which should be discarded as potentially toxic combinations (i.e., trametinib-dinaciclib combination in Patient 11, and two carfilzomib combinations in Patient 12). Not only the effective treatments, but also the predicted doses of drugs in the combinations varied across the patients, indicating that an

optimal balance between treatment efficacy and toxicity should be tailored for each patient.

## Application to ovarian cancer and validation in patient-derived tumor organoids

To investigate whether the prediction approach is applicable also to solid tumors, where large-scale ex vivo drug testing in primary patient cells is more challenging, we employed published scRNA-seq data[25,26] from a cohort of patients with high-grade serous carcinoma (HGSC)[11]. This patient cohort of metastatic tumors with poor responsiveness to standard chemotherapy represents a highly challenging case for personalized treatment identification. We tested the efficacy and

selectivity of the predicted treatments on HGSC patient-derived tumor organoids[27]. Here, three patients were used for the experimental validation of the treatment predictions, with varied sample locations for the scRNA-seq profiling (Patient 1 omentum and Patients 2 and 3 ascites; Suppl. Table 3). To distinguish cancer cells from non-cancerous cells in the organoids and the stromal cell cultures, respectively, we used established tumor marker genes, including *PAX8*, *MUC16* (encoding CA-125) and *EPCAM*, collectively referred to as PAX8+ cells (**Online "Methods"**, Suppl. Figure 5–6).

Due to the small proportion of cancer cell populations detectable in the scRNA-seq profiles of the patient samples (Figs. 3a–c), the identification of cancer subclones for combinatorial targeting was not considered reliable enough in these samples. Therefore, we chose to predict multi-targeting monotherapies, where we used all the cancer cells from the patient sample as a collective malignant entity, disregarding the subclone distinctions. To secure enough fibroblasts and other genetically normal stromal cells for the treatment-selectivity assays, we integrated scRNA-seq data from three HGSC patients (Patients 1, 4 and 5) for the challenging HGSC 1 with an omentum metastasis (Fig. 3a). The expression of PAX8 tumor marker showed a clear separation between the tumor cells and other cell populations also in this "integrated patient" case (Fig. 3a–c, right panels). The treatment-naive tumor organoids were developed exclusively from the cancer cells of the treatment-naïve patient samples (**Online "Methods"**), which displayed an elevated expression of PAX8 (Fig. 3g).

Comparison of the treatment-induced viability changes in the organoid cells and stromal cells in the three patient samples revealed that 31 of the 54 evaluated treatments (57.4%) resulted in greater than 50% inhibition of the PAX8+ tumor cells, and only 11 predicted treatments (20.4%) had similar inhibition levels in the PAX8- non-cancerous cells (Fig. 3d–f, left panels); specifically, the proteasome inhibitors (bortezomib, ixazomib), HSP inhibitor (ganetespib), BET inhibitor (I-BET-762, a.k.a. molibresib), and broad-targeting tyrosine kinase inhibitor (dasatinib) showed notable non-selective responses. Across all the samples and predictions, the patient-specific multi-targeting treatments consistently demonstrated a significantly higher efficacy in suppressing tumor cells, when compared to normal cells (p ≤ 0.01, Wilcoxon test; Fig. 3d–f, right panels). Interestingly, there was no correlation between the predicted treatment doses and PAX8+ or PAX8- cell inhibition effects in any of the patient samples (Suppl. Figure 1b). Similar to the AML patient application, we observed significant differences in the ex vivo drug sensitivities between the patient-specific treatments predicted to be either effective or ineffective when assessed in patient-derived PAX8+ cells (*p* < 0.001, Wilcoxon test; Fig. 3h).

## Landscape of predicted drug responses in solid tumors and hematological cancers

To investigate the versatility and scope of scTherapy in other tumor types, and to study the frequency of recurring therapy options among patients with the same tumor type, as well as the prevalence of predicted personalized treatments unique to a single patient, we used the publicly available scRNA-seq data from cancer patients that were curated and made available by Gavish et al[28]. We expanded our analysis of the AML and HGSC cohorts, and applied scTherapy to three additional tumor types; 10 patients with lung adenocarcinoma (LUAD, 5 primary samples, 5 metastatic, 4 treatment naïve, 6 treated samples)[29], 10 patients with pancreatic ductal adenocarcinoma (PDAC, 5 primary samples, 5 metastatic samples)[30], 4 patients with triple negative breast cancer (TNBC, 4 treatment naïve primary samples)[31]. These cancer types were chosen as they pose clinically significant therapeutic challenges; in particular, patients with PDAC, LUAD and TNBC urgently need new treatment options. For comparison, we included 12 samples from our AML cohort reported in this manuscript, and 4 ovarian

samples from the HGSC cohort of metastatic tumors with poor responsiveness to standard chemotherapy, representing a highly challenging case for personalized treatment identification.

Our analysis demonstrates the presence of therapy clusters shared among patients with the same tumor type, as well as emergence of unique, patient-specific therapies (Fig. 4a). These findings provide a statistical landscape of the predicted treatments: 19% are patient-specific, indicating a high degree of personalized treatment potential, 25% are disease-specific (2% LUAD, 1% TNBC, 2% PDAC, 10% AML, 10% HGSC), and 22% were common across the five cancer types (Fig. 4b). While we observed a relatively clear separation between solid tumors and hematological cancers, as expected, there were no apparent therapy clusters for specific diseases stages, or striking differences between treatment naïve and treated patient samples (Fig. 4a). The predicted treatments for the solid tumors were also rather equally distributed (Fig. 4c). This distribution underscores the versatility of scTherapy in addressing the diverse and complex landscape of cancer treatment predictions, paving the way for identification of more targeted and effective therapeutic strategies across a range of tumor types. Furthermore, 22 out of the total 131 scTherapy-predicted treatments (17%) are currently advancing through phase 3 or 4 clinical trials (Suppl. Table 4), underscoring the capability of the prediction approach to identify clinically relevant and potentially efficacious cancer treatment options.

## Quantitative comparison of scTherapy against state-of-the-art methods in the field

We compared the patient-specific predictions of scTherapy against those of BeyondCell[15] and scDrug[17] in the AML patients, where large-scale single-drug sensitivity testing is routinely done for each patient sample. We focused on the top and bottom 15 drugs predicted as the most and least effective, respectively, by each method for the individual patients. To ensure a fair comparison with other methods, which do not offer effective dose predictions like scTherapy, we focused on the overall drug responses across the dose ranges, and summarized the drug efficacy with the drug sensitivity score (DSS, normalized area under the drug dose-response curve), which is widely-used in personalized drug testing studies[32]. We observed a consistently improved performance of scTherapy for prediction of both effective and ineffective single-drugs across the patients, when compared to BeyondCell and scDrug (Fig. 5a, b)

To summarize these quantitative evaluations, we used the Receiver Operating Characteristic (ROC) curves and calculated Area Under the ROC Curves (AUC) values for each prediction method. The ROC curves visualize the overall performance of the methods to discriminate between effective and ineffective drug treatments, using the experimentally measured DSS values aggregated across the 12 AML patients (Fig. 5c). The AUC values and confidence intervals clearly demonstrate the superior predictive performance of scTherapy (Fig. 5d). The improvement in predictive accuracy is statistically significant, as evidenced by DeLong test, where the performance of scTherapy was significantly better compared to that of both scDrug and BeyondCell (p < 0.01). These quantitative and statistical evaluations provide further support for the predictive accuracy of scTherapy, in addition to the experimental validations.

## Discussion

Advanced cancers are heterogeneous diseases, typically comprising at diagnosis more than $10^{10}$ cells, which very likely harbor therapy-resistant subpopulations[11,24]. This translates into a medical need for multi-targeting therapies for effective cancer cures. Our experimental-computational approach for personalized identification of multi-targeting treatments makes use of two recent advances: (i) the feasibility of scRNA-seq profiling in complex patient samples that allows for the identification of malignant and non-cancerous cell populations for

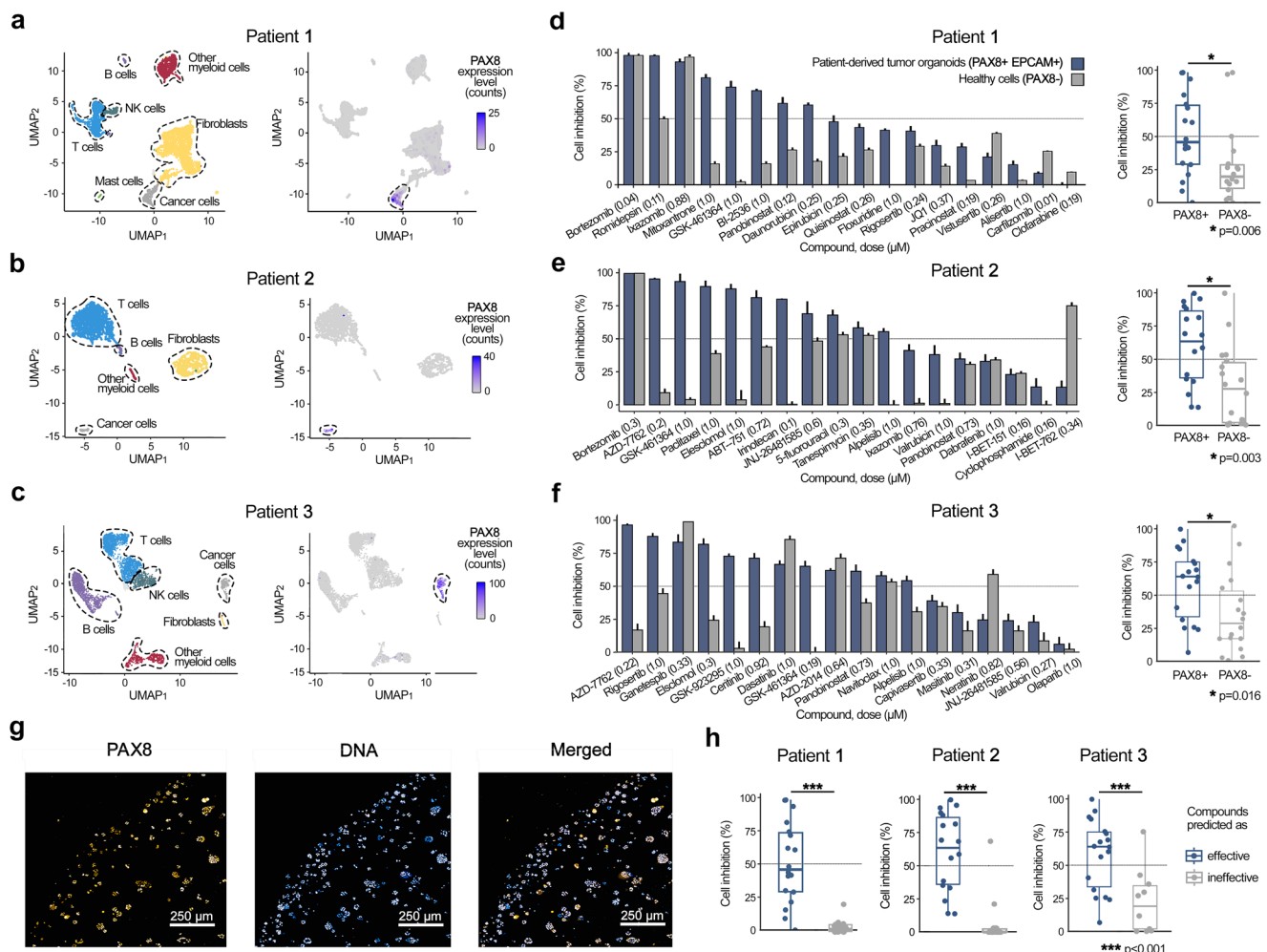

**Fig. 3 | Experimental validation in ovarian cancer patient-derived tumor organoids. a–c** UMAP projection of the scRNA-seq transcriptomic profiles for the three HGSC patient samples, using standard Seurat integration workflow, where cell types were identified with ScType (left panel). The number of cells analyzed per patient as shown on UMAP plots: Patient 1 (4706 cells), Patient 2 (1483 cells), and Patient 3 (1934 cells). Expression of the PAX8 marker, effectively separating tumor cells from the other cell populations (right panels). Scale bars represent raw PAX8 expression counts. **d–f** Barplots showing cell inhibition differences between the patient-derived organoid cancer cells (PAX8 + , blue bars) and non-cancerous normal cells (PAX8-, gray bars) for the 18 predicted multi-targeting drugs (left panels). The predicted effective doses are indicated in parentheses (µM), and the dotted vertical lines indicate 50% inhibition. The error bars represent SEM, based on three replicates of organoid treatments and curve-fitting in PAX8- cells. For patients 2 and 3 both organoids and the stromal cell cultures were available at the cell numbers sufficient for single-drug sensitivity and selectivity testing, whereas

for Patient 1 the PAX8+ tumor cells originated from the patient organoid and PAX8-normal cells were available from additional Patients 4 and 5 (Suppl. Table 3). Statistical comparison of the treatment responses between PAX8+ and PAX8- cells across three HGSC patient samples with two-sided Wilcoxon signed-rank test (right panels). **g** Representative immunofluorescent image of treatment-naive tumor organoids from Patient 1 sample. This experiment was conducted once. Scale bar equals to 250 µm. **h** Statistical comparison of ex vivo drug sensitivity differences in patient-derived PAX8+ cells between the treatments predicted by scTherapy to be either effective ($n = 18$) or ineffective ($n = 10$) in the individual patients ($p < 0.001$, two-sided Wilcoxon test). Suppl. Fig. 7 shows the same data in a heatmap, summarizing the drug responses across multiple doses using drug sensitivity scores (DSS), instead of the percentage inhibition at the predicted effective dose (as shown here). Box plots show the median (central line), 25th and 75th percentiles (box edges), and the range within 1.5 times the interquartile range from the box (whiskers). Source data are provided as a Source Data file.

selective targeting; and (ii) the availability of large-scale transcriptomic and viability response profiles of cancer cell lines treated with thousands of single-agent perturbations. Taken together, our approach provides a clinically actionable and relatively fast means for predicting drug-dose combinations for individual patients, and compared to our earlier work[33], it can be applied also for patients whose tumors are not easily amenable to drug testing (e.g. HGSC). The only input for the model is a count scRNA-seq data matrix of a given patient sample; the rest of the computational steps are either fully-automated or semi-automated (e.g., selection of the broad-level subclones based on visual analysis of the clonal evolutionary tree; see Step 4 in Suppl. Figure 3). To validate this targeting strategy, we demonstrated in AML and HGSC cases that nearly all the predicted combinations exhibited positive synergy scores (96.3%), highlighting their potential for improved

therapeutic efficacy and reduced toxicity by lowering the doses of single agents. Importantly, 83.4% of the predictions demonstrated low-toxicity to normal cells (< 50% inhibition of non-cancerous cells); however, the flow cytometry and organoid drug response assays indicated that certain multi-target therapies (16.6%) excessively inhibited non-cancerous cells (e.g. proteasome and topoisomerase inhibitors), emphasizing the importance of ex vivo experimental validation prior to clinical translation.

scTherapy identifies individual drugs or their combinations that (i) reverse clone-specific transcriptomic responses closer to the normal expression state, and (ii) exhibit selective cancer cell inhibition at the predicted effective dose to ensure differential inhibition between malignant and normal cells. The model outcome is a list of predicted treatments and effective doses for targeting the unique intratumoral

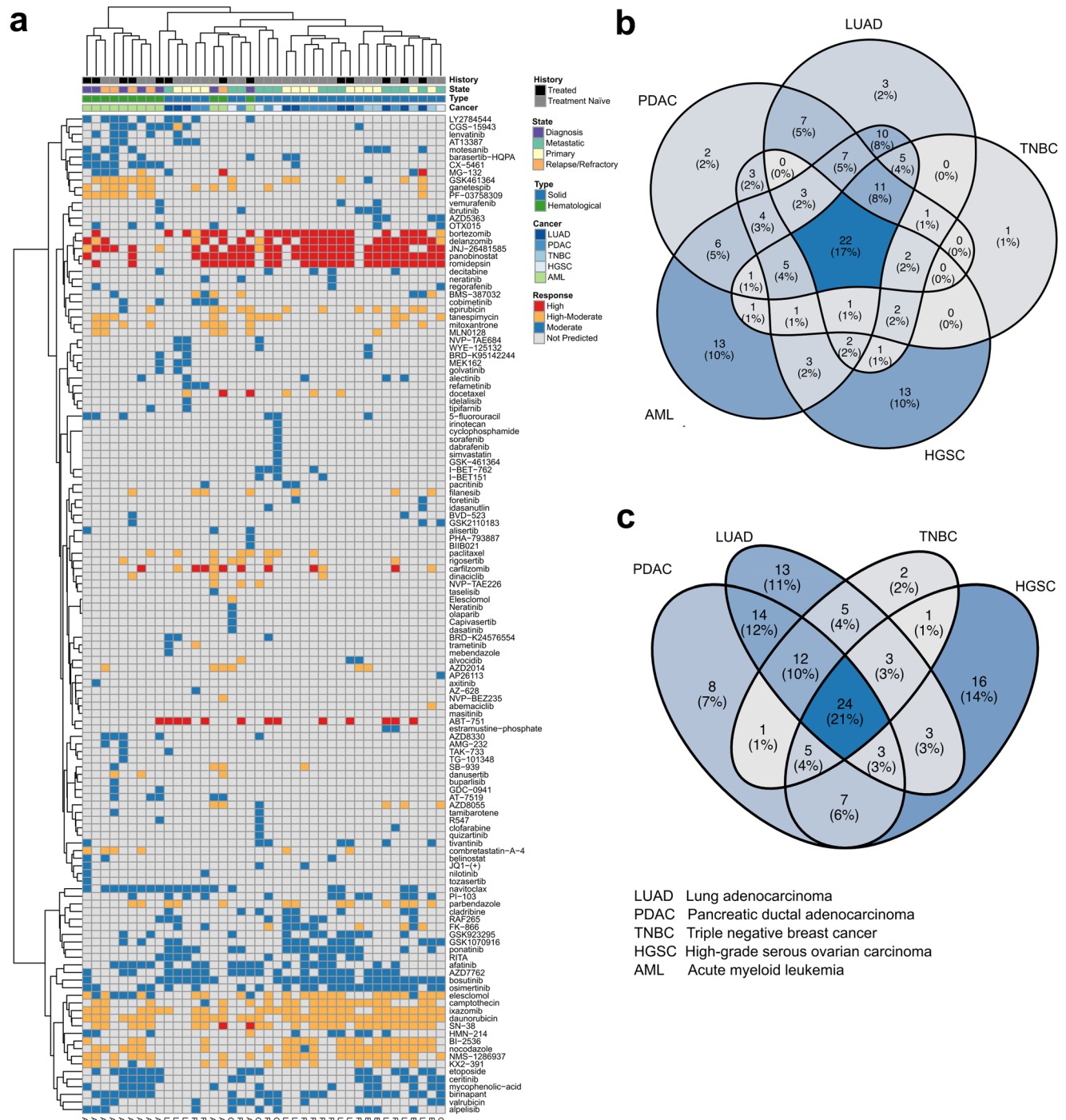

**Fig. 4 | scTherapy predictions across solid tumors and hematological cancers.**
**a** Heatmap of the scTherapy predicted monotherapies across patient samples from multiple tumor types (columns). Each column represents a patient sample, with annotations indicating treatment history, disease stage, and cancer type. The color coding indicates predicted effectiveness of the treatment (rows). Instances where the ScTherapy did not generate a prediction for a specific patient sample are marked with a gray color. **b** Overlap of the predicted treatments between patients with solid tumor (PDAC, TNBC, LUAD, HGSC) and patients with hematological cancer (AML) (**c**). Overlap of the predicted treatments when focusing on the solid tumors only (PDAC, TNBC, LUAD, HGSC). Source data are provided as a Source Data file.

heterogeneity within each patient sample, complemented with a confidence score for the reliability of each treatment-dose prediction. The quantitative performance evaluation (repeated cross-validation and experimental validations), together with the confidence scoring (conformal prediction), enables medical professionals to decide when and how to use the model to guide clinical decision making. By mapping the gene signatures to drug-target interactions networks, one can also explore potential biomarkers (e.g., patient-specific DEGs) that

drive the selection of the best treatment regimens for individual patients (Suppl. Fig. 8). This provides additional insights into the rationale of the treatment recommendations for a given patient. Such network markers are not limited by the current genetic biomarkers, e.g., oncogenes, that are rare especially for drug combinations[34]. scTherapy can also predict responses to custom compounds, hence facilitating the assessment of novel or less-studied compounds for their patient-specific efficacy. Furthermore, the model incorporates a

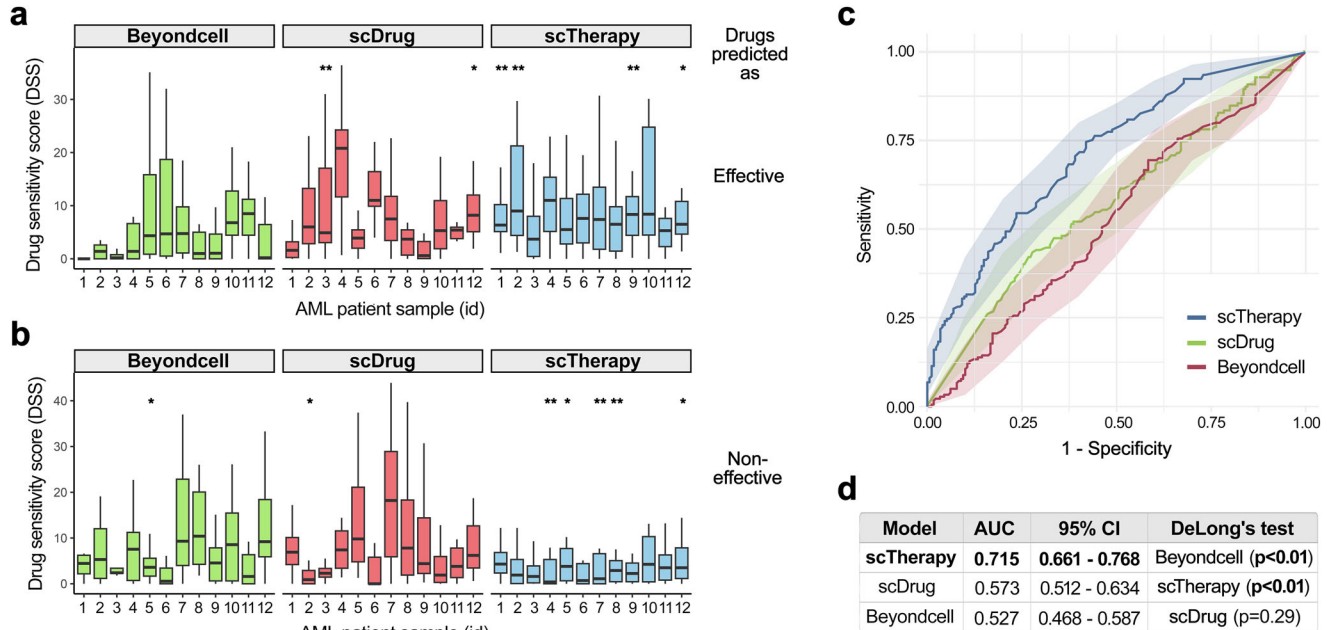

**Fig. 5 | Quantitate comparison of monotherapy efficacy predictions in 12 AML patients.** Drug sensitivity score (DSS)[32] distributions of the top-15 drugs predicted as (**a**) the most effective and (**b**) the least effective monotherapies by each model for individual patients (n = 12). For each patient, 15 technical replicates (top-15 drugs) were compared between models using two-sided pairwise Wilcoxon rank-sum tests, with p-values adjusted for the False Discovery Rate (FDR) with the Benjamini-Hochberg procedure. *a model's predictions are significantly different (p < 0.05) compared to that of at least one of the two other methods; **a model's predictions show a significant difference (p < 0.05) when compared to both of the alternative methods. Box plots show the median (central line), 25th and 75th percentiles (box edges), and the range within 1.5 times the interquartile range from the box (whiskers). **c** Receiver Operating Characteristic (ROC) curves for each model demonstrating their ability to distinguish between effective and ineffective treatments based on the predictions of the most effective and least effective drugs by each model for 12 individual patients (as shown in panels **a** and **b**). The shaded area around each curve represents the 95% confidence interval (CI), calculated around the mean, illustrating the variability of the predictions based on data from 12 patients. **d** A summary table displays the Area Under the Curve (AUC) values for each model, quantifying their overall predictive accuracy. Statistical comparison was performed with the two-sided DeLong's test, indicating that the prediction performance of scTherapy was significantly better than that of scDrug and BeyondCell (p = 0.00064 and p = 0.0000054). The other two methods show statistically similar prediction results (p = 0.29). Source data are provided as a Source Data file.

user-defined drug-dose information, especially useful in cases where certain drugs or doses are clinically more relevant for a given cancer type. By dose restriction, one can further reduce the risk of toxic effects that often occur at higher doses, hence making the predictions clinically more feasible. Overall, our approach offers a systematic and flexible framework for predicting personalized drug-dose combinations that can be tailored to individual patient and tumor characteristics.

When comparing predictions across five cancer types, we observed a rather striking balance between treatment options that are recurrent in the patients of the same tumor type (25%), in addition to a strong inclination towards personalized options (19% of treatments being patient-specific), and high prevalence of common treatment predictions between cancers even among targeted therapies (22%) (Fig. 4). When comparing solid tumors and hematological cancers, we predicted a higher number of targeted signal transduction inhibitors for the AML patients, compared to the patients with HGSC, which reflects the underlying differences in the disease biology. AML cells often carry oncogenic mutations in signaling proteins, making the cells addicted to MAPK signaling[35], which explains why MEK inhibitor combinations were identified for many of the AML patients. Similarly, PLK inhibition has been extensively studied in AML, and while PLK inhibitor combinations have shown promise in clinical development, they are also associated with complicated toxicities[36]. Therefore, even though the scTherapy model identifies surprising multi-targeting treatments for patients, the drug and target classes of the predicted combinations are well-studied in these cancer types, and many are either approved or under clinical investigation (Suppl. Table 4). Importantly, the model predictions are tailored to the molecular

context of a given patient (or sample), which is expected to lead to better efficacy-safety balance at the level of an individual patient, rather than identifying broadly chemotoxic combinations that may lead to severe side effects in the non-matching subset of patients. Our next-generation precision medicine approach provides a streamlined, yet relatively precise approach to finding the right combinations of drugs and doses, toward enhancing therapeutic potential through using both molecular and functional information.

Traditionally, effective drug combinations have been identified either by empirical clinical testing[37], or using high-throughput screening (HTS) in cell line panels in vitro, followed by in vivo validation of the most relevant combinations and target mechanisms in animal models[38-40]. However, drug combination synergy is a rare and highly context-dependent event, which requires combinations to be tested in large-scale screens and in various cellular contexts and genomic backgrounds[34]. This is beyond the scalability of in vivo models, and in vitro screening alone cannot identify combinations targeting specific cancer subclones, even if large enough cell line panels can to certain extent model the cellular heterogeneity and drug response variability. In particular, a multi-targeting therapy that effectively inhibits cancer cells may also co-inhibit normal cells, rendering the treatment non-selective against malignant cells. In patient applications, it is therefore critical to identify cancer-selective combinations, rather than broadly active therapies that may lead to severe toxic effects. Ex vivo drug testing in primary patient cells, using either patient-derived 2D cell cultures or 3D organoids, strikes a balance between the in vitro and in vivo approaches[21,41]. However, even though flow cytometry and imaging-based ex vivo assays offer possibilities for drug response testing at a single-cell resolution, HTS of a larger

number of drug combinations in multiple doses remains infeasible in scarce patient cells using these advanced assays[42–44]. Therefore, systematic methods to prioritize the most potential combinations to be tested in primary patient cells are needed.

Various machine learning (ML) methods have been developed to predict effective anticancer drug combinations using multi-omics training data from large-scale screens in cancer cell lines and patient-derived samples. By surveying the existing ML methods[45], we identified three critical areas of improvement for translational applications. First, none of the existing methods were designed to predict selective drug combinations that target multiple cancer subclones, and avoid co-inhibiting normal cells, using merely single-cell transcriptomic data as input. This is important since multi-omics profiling and ex vivo drug testing in scarce primary patient cells is not practically feasible for many tumor types[21]. Second, most of the current methods either do not use any normal reference, and hence lack preclinical toxicity predictions, or use molecular or functional profiles from healthy individuals to de-prioritize toxic combinations, which may lead to non-selective combination predictions, due to high inter-individual molecular and phenotypic heterogeneity. Third, drug combination effects are not only patient-specific, but also highly dose-dependent[34], and therefore we argue that computational prediction methods need to provide dose-specific prediction of the responses, especially for translational applications where often the lower doses are better tolerated by the patients. However, we note that doses optimized in cell-based assays need to be adjusted for clinical use, as patient treatment doses depend on multiple factors, including age, disease stage and comorbidities.

Similarly, a recent review of single-cell level drug response prediction methods concluded three key limitations of the current methods[20]: (i) they cannot predict drug combination responses, but rather focus on single drug effects; (ii) they do not attempt to predict effective drug dosage, which is critical for translational applications; and (iii) they have not been experimentally evaluated to truly evaluate their practical utility. For instance, BeyondCell introduced a "therapeutic cluster" (TC) concept, defined as groups of cells with a similar drug response within cellular populations, and applied it to propose single-drugs to target sensitive and resistant cells and to identify drug-response biomarkers[15]. In BeyondCell, TCs were identified using unsupervised clustering analysis, whereas scTherapy takes a more supervised approach to identifying multi-targeting treatment regimens that inhibit multiple disease- or resistance-driving cancer clones in individual patients. Similarly, scDrug starts by cell clustering of scRNA-seq data set and identifies tumor cell subpopulations, and then uses functional annotation of subclusters to suggest candidate drugs for effective treatments[17]. Our work addresses all these limitations of the current methods for single-cell level drug treatment response prediction in cancer applications. When compared with the measured single-agent responses in the AML patients, we showed that Beyond-Cell or scDrug predictions provided sub-optimal prediction accuracy compared to scTherapy patient-specific predictions (Fig. 5).

The current version of ScTherapy requires only the scRNA-seq count matrix of a cancer sample as its input. This means that tumor types or patient cases in which the most predictive omics feature for a given targeted therapy is a point mutation are currently beyond the scTherapy approach; for instance, melanomas harboring BRAF-V600E mutation are known to benefit from BRAF inhibitors such as vemurafenib. In future developments, we plan to incorporate multi-omics data, including point mutations, where available. We further utilized CNV profiles inferred from scRNA-seq data (using InferCNV), rather than the actual CNV measurements, to streamline the methodology by requiring only scRNA-seq data from patient samples as input for scTherapy. Previous studies have shown that inferred CNVs can accurately capture the complex subclonal architecture of tumors based on their genomic differences[46–48]. We also compared scRNA-seq-

inferred CNV profiles with the expression-based cell clusters in the patient 12, used for combinatorial testing, and showed that expression-based clustering may miss certain subclonal genomic variation in the detected cancer subpopulations (Suppl. Fig. 9). However, while this study focused on the differential drug targeting between genetic cancer clones and non-cancerous cells, the predictive approach is applicable also to selective targeting of other cell types, states or phenotypic subpopulations, for instance, small molecule-induced activation of immune cell TME compartments for boosting treatment effects of targeted-drugs or immunotherapies. Additionally, with the emerging availability of large-scale morphological and single-cell proteomic response profiles in cancer cell lines and samples[49–52], future scTherapy versions could be extended to incorporate these phenotypic measurements for better modeling and targeting of different mechanisms of drug resistance and sensitivity.

In samples where scRNA-seq detects only a small fraction of cancer cells, further dividing this subset becomes unreliable, as it would increase noise rather than distinguishing cancer-related signals. Given this challenge in our HGSC samples, we chose to predict responses to single-agents, instead of drug combinations, relying on a comprehensive transcriptomic profile of the cancer cells (instead of smaller sub-clones). Therefore, we advise users of scTherapy to consider the limitations of cell population size and the potential for increased noise when attempting to delineate subclonal populations. In such cases, the analysis of the overall transcriptomic landscape of malignant cells may offer a more feasible and effective strategy for identifying multi-targeted monotherapies, rather than subclone-specific drug combinations.

However, when using scTherapy for predicting combination therapies, such as in the AML case, one can shift focus to broader level subclones. While our current predictions are made for two major subclones, where a specific drug in the drug combination is uniquely predicted for each subclone, these major subclones themselves encompass a spectrum of subclonal diversity. In future developments, we plan to extend scTherapy for multi-drug combinations that can also selectively target other minor subclones, e.g., three or more subclones, in cases where this is feasible. However, we note that the current predictions from scTherapy can also target both major and minor subclones, due to polypharmacological effects of the drugs. For instance, when examining the drug predictions among various subclones in the AML patient samples, we observed that the treatment predictions for the major subclones largely coincide with those of minor subclones within the same broad clone (Suppl. Fig. 10). This demonstrates a broad yet selective targeting strategy across the clonal spectrum of cancer cells.

Compounds from different drug and target classes may elicit varied phenotypic responses in the viability and transcriptional response profiles. For instance, in contrast to other molecularly-targeted compounds, HDAC inhibitors often induce significant changes in the expression of multiple genes beyond their target proteins. Comparison of the expression and viability changes between cancerous and non-cancerous cells is expected to normalize out a part of such variability between drug classes. However, future studies are warranted to tailor input data not only for a specific patient sample but also make predictions drug class-specific by considering differences in binding affinities, phenotypic profiles, and treatment time points. Different disease models may also have differing growth dynamics. For instance, as opposed to the most conventional cell lines, organoid cells undergo less cell divisions during 7-day incubation. Therefore, some of the discrepancies seen between the model predictions (made using in vitro cell line data) and the experimental validations (made in ex vivo experiments) may stem from such variations between the 2D and 3D disease models and time points. The ex vivo validation strategy for the selective predictions would also benefit from the incorporation of different types of control models, ideally closely matching the growth

conditions of the cancer cells, which would minimize confounding factors, such as medium composition, matrix requirements, or growth dynamics of the model. In this context, adding different types of "healthy controls", for instance, orthotopic non-transformed cells together with PBMC or iPSC organoids from the same patient, would offer alternative validation strategies in solid tumors.

## Methods

This research complies with all relevant ethical regulations, approved by the institutional review boards, who approved the use of the human samples in the study. The AML patient samples and data were collected and published with signed informed consent in accordance with the Declaration of Helsinki (HUS Ethical Committee Statement 303/13/03/01/2011, latest amendment 7 dated June 15, 2016. Latest HUS study permit HUS/395/2018 dated February 13, 2018). The HGSC patient samples were collected as a part of a larger study cohort, where all patients participating in the study provided written informed consent. The study and the use of all clinical material have been approved by The Ethics Committee of the Hospital District of Southwest Finland (ETMK) under decision number EMTK: 145/1801/2015.

### Compiling a large-scale phenotypic response data for pre-training a LightGBM model

A comprehensive training dataset of large-scale phenotypic response profiles was created by merging data from three databases: Connectivity Map LINCS 2020[22], PharmacoDB[23], and PubChem[53] (Suppl. Figure 3, bottom part). These continuously expanding, publicly available databases allowed us to establish an extensive dataset that provides functional information on both viability and transcriptomic responses to increasing numbers of compounds. Details on the dataset used in the present study are outlined below. The Connectivity Map (CMap) LINCS 2020 is a reference database that houses gene expression response profiles of 12,328 genes measured in 240 cell lines across multiple doses and time points for 39,321 small-molecule compounds. Additionally, LINCS 2020 data includes paired control states for each perturbagen-cell line combination, enabling a comparison of the transcriptional changes before and after each treatment. To supplement our dataset, we leveraged information from PharmacoDB, a database that contains dose-response viability data for 56,149 drugs across 1758 cancer cell lines at multiple doses. For further analysis, we employed 10,303 overlapping compound-cell line pairs, which were common between 24 h transcriptional responses from CMap LINCS 2020 (passing quality control, i.e., qc_pass = 1) and PharmacoDB. For matching compounds between PharmacoDB and CMap LINCS 2020, we used compound identifiers, and for the cell line matching, we used cellosaurus IDs[54]. To extract structural information of the compounds, we used PubChem and RDKit (rcdk v3.6 and rcdklibs v2.3) to generate molecular fingerprints (ECFP4) from the SMILES representation of each common drug[55].

The light gradient boosting machine (LightGBM) model was trained on a comprehensive dataset of 3695 compounds tested at 1–35 doses in 167 cell lines. Drug-dose-cell line profiles (including transcriptomic response profiles, ECFP4 molecular fingerprints, and drug doses) were used as the model predictors, while the outcome variable is the inhibition percentage, derived from PharmacoDB dose-response viability data (Suppl. Figure 3). The LightGBM model was trained using Bayesian Optimization, with a repeated cross-validation (three repetitions), and ten-fold inner cross-validation (CV). This ensures a robust and generalizable model for patient applications. More specifically, the LightGBM model matches gene expression signatures (differentially expressed genes between cancer and non-cancer cells) to the transcriptional responses to small molecules tested at different doses from LINCS 2020 to find the compounds that induce opposite transcriptomic changes. In the next step, the model identifies which compounds and doses most effectively inhibit cell growth, by

extracting percent inhibition responses for corresponding cell line-drug-dose triads from PharmacoDB. After examining tens of thousands of possible matches, the model provides a prediction of the most promising compounds and the effective dose. We also recommend including at least one dose-fold above and below the predicted dose in the experimental evaluation to delineate the most effective and least toxic drug dosage.

### Prediction of multi-targeting therapies using scRNA-seq data in AML patient samples

The experimental-computational prediction approach consists of the following five subsequent steps (Suppl. Figure 3). These steps are described here for the AML case, and modifications to this pipeline in the HGSC case are described under section **Tailoring the experimental-computational approach to ovarian tumor patient samples**.

#### Step 1: Longitudinal sampling

After obtaining informed consent, bone marrow aspirates were collected from patients diagnosed with acute myeloid leukemia (AML) at the Helsinki University Hospital (HUS). For this study, a total of 12 longitudinal samples (7 at diagnosis, 2 at relapse stage and 3 at refractory stage) were obtained and stored at the Finnish Hematology Registry and Clinical Biobank (FHRB). The protocols used for this study were reviewed and approved by the institutional review board in compliance with the Declaration of Helsinki[56]. The below steps 2–5 were repeated for each sample individually to provide a customized set of effective and low-toxic multi-targeting options for each patient individually by considering the intratumoral heterogeneity of cancer cells that is present not only at later stages of the disease or resistance development but already at the diagnostic stage.

#### Step 2: Single-cell data analysis

For the single-cell transcriptomic analysis, we processed the filtered gene-barcode matrix derived from 10X Genomics data using the ScType platform[57], with Louvain clustering, as implemented in the Seurat version 4.3.0[58]. To filter out low-quality cells, we removed cells that had either a low or high number of detected genes and also cells that had more than 10% of mitochondrial UMI counts in the AML scRNA-seq data. The quality control (QC) criteria depend on the sample types; for instance, in HGSC organoids, 20% of mitochondrial UMI count cut-off was used[27]. Such QC cell filtering step is critical to exclude technical noise and thus to avoid biases in the downstream analysis. To normalize the gene expression levels, we utilized the LogNormalize method implemented in Seurat.

#### Step 3: Identification of malignant and normal cells

Single-cell RNA sequencing profiles were used to identify malignant and normal cell clusters in each sample using three analytical tools, ScType[57], CopyKAT[31], and SCEVAN[59]. These tools were specifically selected for their ability to accurately classify and differentiate between malignant and normal cells in the given complex sample, eliminating the requirement for larger cohort samples. We demonstrated that the detection procedure maintains a surprisingly stable performance in most AML samples, even when as large proportion as 75% of cells are removed (Suppl. Fig. 11). This suggests its robust performance on diverse datasets with different proportions of healthy and malignant cells, and relatively stable capability to capture relevant malignant and healthy signatures of mixed cell samples.

#### Step 3a: Cell type annotation

We utilized the ScType web-tool[57] that enables fast, precise and fully-automated cell cluster annotation. ScType integrates cell type markers from the two most comprehensive resources for human cell populations and classifies cells based on gene expression changes across

clusters. We used ScType to assign a confidence score to each cell type annotation and each cluster, with high scores indicating a high level of confidence in the cell type annotation. Clusters with low scores were labeled as "Unknown" cell types based on the default ScType cutoff (score < number of cells in the cluster divided by 4). In addition, we visually analyzed previously established marker genes for blasts, including CD33, CD34, CD38, PROM1, ENG, CD99 and KIT[33], on the UMAP space and calculated the proportion of the blast cells in each patient sample to gain a better understanding of the distribution of leukemic cells. This resulted in a Seurat object that includes cell clusters and their corresponding annotations.

## Step 3b: Detection of aneuploid cells

To further classify cell populations as normal or malignant, we developed an ensemble approach that utilizes multiple methods to generate a confident classification. The first method is a marker-based approach, which involves carefully filtered cell markers from Cell-Marker2.0 database[60], and then using these as a custom marker dataset for ScType to identify normal and malignant cells. The second approach uses CopyKAT[31], a Bayesian segmentation-based method, with default parameters and known normal cells (T cells in the AML case[61]) as a baseline to estimate copy number alterations (CNA). The third method is SCEVAN, with the non-cancerous control cells used as input, which employs a Mumford and Shah energy model to distinguish normal and malignant cell states[59]. The use of CNA estimation-based approaches allows us to classify malignant cells while taking into account overall variability within normal cells. We then constructed a majority vote based on the combined results of these tools to confidently identify both normal and malignant cell clusters. To further validate our approach, we superimposed the ensemble predictions onto the UMAP space and compared them with the cell-type information obtained from ScType. By integrating cell type and normal/malignant annotations from ScType, with ploidy information from CopyKAT and SCEVAN, we identified clusters of cells as either normal or malignant. Our ensemble approach accounts for variability within normal cells and therefore minimizes the risk of misclassification.

## Step 4: Identification of genetically distinct subclones and visualizing clonal lineages

After successfully identifying normal and malignant cell clusters, we used inferCNV[62] to infer large-scale copy number variations, such as gains or deletions of whole chromosomes or segments from the scRNA-seq data. The input for the inferCNV analysis included the known non-cancerous cells identified in Step 3, genomic locations, cell type annotations, and the scRNA-seq count matrix data. CNVs were inferred using the Hidden Markov Model (HMM) approach implemented in the 6-state i6 HMM model (https://github.com/broadinstitute/infercnvApp/blob/master/inst/shiny/www/Infercnv-i6-HMM-type.md). In accordance with the inferCVN guidelines in the document "Using 10X data" section (https://github.com/broadinstitute/infercnv/wiki/infercnv-10x), we adjusted the "cutoff" parameter from 1 to 0.1, and subsequently computed the CNV profiles from the scRNA-seq expression counts. To explore the subclonal structures, we used the "subcluster" method on the HMM predicted CNVs.

After identifying the genetically distinct subclones, we used Uphyloplot2[63] to visualize intra-tumoral heterogeneity and clonal evolution using the CNV calls from the inferCNV 6-state HMM "subcluster" method and its ".cellgroupings" file. We note that the resultant evolutionary tree does not follow a molecular clock; rather, the branch length is proportional to the percentage of cells in the subclone, hence providing information about which subclone dominates the tumor mass. Next, two broad-level subclones detected from the evolutionary tree were identified using visual analysis, and along with normal cells, overlaid on a UMAP projection for further analysis. To quantify gene expression differences between the normal cells (identified in Step 3)

and the broad-level subclones (identified in Step 4), log-fold change values and determined significance levels via the nonparametric Wilcoxon rank-sum test, applied in Seurat 4.3.0 using the FindMarkers command.

## Step 5: Predictive modeling of multi-targeting therapies

When applied to patient samples, the subclone-specific differentially expressed genes (DEGs) were used as input for the pre-trained LightGBM model to predict single-agent cell inhibition percentages for each compound-dose pair in the particular patient cells. This allows us to take into account both the intratumoral and intertumoral heterogeneity, as captured by the scRNA-seq profiles of the patient samples. Our prediction approach is highly flexible and can be used in two ways: first, by utilizing a predefined set of drug-dose pairs for predictions, or second, by customizing the analysis with additional input of new drug structures (ECFP4 fingerprints) and/or specific doses of interest. The scTherapy tool offers flexibility for the users in performing either monotherapy or combination therapy predictions, depending on the number of cancer cells available from the patient samples for experimental testing. Currently, the predictions are limited to two majority clones, where a specific drug in the drug combination is uniquely predicted for each subclone, when the model is used for combinatorial treatment predictions (please see the ovarian cancer case study below for monotherapy predictions).

As any ML model predictions inherently come with some degree of uncertainty, we used conformal prediction (CP) to eliminate low-confidence predictions and improve the prediction accuracy[64]. CP generates confidence intervals for each prediction by measuring uncertainty based on repeated CV residuals. Predictions with a non-conformity score < 0.8 were excluded, thereby ensuring inclusion of only confident and accurate predictions. In addition, to ensure that our model returns clinically more relevant predictions, we imposed a 1 µM dose maximum when utilizing the pre-defined set of drug-dose pairs. High drug doses, even though potentially increasing cancer cell inhibition, may also inhibit normal cells, hence compromising the selectivity of targeted agents[65]. By using such a dose restriction, we ensured the selectivity of targeted agents returned by the model and minimized the risk of toxic effects, making our predictions more clinically actionable. We applied this approach to each subclone, hence generating a set of drug-dose-response tuples for the experimental validation.

## Retrospective testing of the model predictions in single-agent data from AML patients

To validate the performance of our model, we first used existing data from bulk drug response assays, available for the 12 patient samples from previous studies[56]. For the single-agent response testing, 20 µl of fresh AML cell (approximately 10,000) suspension in mononuclear cell medium was added per well to pre-drugged plates with 10-fold dilution series of five concentrations, and the whole-well cell viability was measured with CellTiter-Glo (CTG; Promega) in duplicate using established protocols[35,56]. After 72 h of incubation at 37 °C and 5% CO2, cell viability of each well was measured using the CTG luminescent assay and a PHERAstar FS (BMG Labtech) plate reader. The percentage inhibition was calculated by normalizing the cell viability to negative control wells containing 0.1% dimethyl sulfoxide (DMSO), and positive control wells containing 100 µM cell killing benzethonium chloride (BzCl). Notably, these existing single-agent response data were not used in the model training and were only employed retrospectively to test the accuracy of the model to predict effective monotherapies. Since the whole-well assay is not a cell population-specific assay, we performed this validation using the differentially expressed genes (DEGs) between the malignant cell types and normal cells to generate single-agent predictions for each patient sample. Subsequently, we matched the drugs and doses predicted by the model to the available patient-specific cell viability dose-response data (see Fig. 2b).

## Prospective testing using whole-well and flow cytometry assays in the AML patient cells

The patient-specific predicted combinations were first tested on the bone marrow mononuclear cells of each patient in a 4 × 4 dose-response matrix using the bulk CTG viability assay, similarly as before[33]. The combination synergy in the experimental validations was quantified using ZIP model[66], calculated based on the dose region around the predicted effective dose of each compound in the combination.

Cell population-specific drug combination effects in primary AML patient samples were assessed by high-throughput flow cytometry assay. The compounds were dissolved in 100% dimethyl sulfoxide and dispensed on conical bottom 384-well plates (Greiner) either as single agents or combinations using an Echo 650 liquid handler (Beckman Colter). Cryopreserved bone marrow mononuclear cells were thawed and suspended in 12.5% HS-5 derived conditioned medium, and $2-3 \times 10^4$ live cells were seeded with a MultiFlo FX.RAD (BioTek) to 384 well-plates, followed by incubation for 72 h at 37 °C and 5% $CO_2$. To profile the cell sub-population responses, the cells were stained with BV785 Mouse Anti-Human CD14 (Biolegend, dilution 1:200), VB515 Recombinant Anti-Human CD56 (Miltenyi, dilution 1:400), and following antibodies from BD Biosciences; V500 Mouse Anti-Human CD45 (dilution 1:240), BV650 Mouse Anti-Human CD19 (dilution 1:120), PE-Cy7 Mouse Anti-Human CD3 (dilution 1:150), PE Mouse Anti-Human CD34 (dilution 1:240), BV421 Mouse Anti-Human CD38 (dilution 1:600) and APC Mouse Anti-Human CD117 (1:600), together with APC-Fire 750 Annexin V (Biolegend, dilution 1:80) and DRAQ7 (BD Biosciences, dilution 1:600). The cells were analyzed with an iQue3 flow cytometer (Sartorius). The remaining live cells after drug treatments were gated using Forecyt (Sartorius). Briefly, cell singlets were identified based on FSC-A (forward-scattered area) versus FSC-H ratio, and live cells were identified by excluding annexin V- and DRAQ7-positive cells, followed by identification of leukocytes (CD45 + ). Further characterization was done for NK cells (CD56 + CD3-), leukemic blasts (CD34+ and/or CD117 + ) leukemic stem cells (CD34 + CD38-), monocytes (CD14) and T/B- cells (SSC-A and CD3/19) from the leukocytes.

## Tailoring the experimental-computational approach to ovarian tumor patient samples

To differentiate between cancer and non-cancerous cells in ovarian cancer patient scRNA-seq data, we utilized a panel of established marker genes, including PAX8, CA125, MUC16, WFDC2, and EPCAM, collectively referred to as PAX8+ cells; PAX8 is expressed in 80–96% of high-grade serous ovarian cancer (HGSC) tumors (Suppl. Figure 5)[67,68]. Our initial analysis focused on the HGSC Patient 1 sample, selected due to the availability of both scRNA-seq data and viable cells for experimental validation. Due to the small proportion of PAX8 + malignant cells detected in the scRNA-seq data, we opted to predict only single-agent therapies as opposed to combination therapies. However, during the validation phase, the PAX8- stromal cells of Patient 1, serving as normal controls, died. This led us to integrate this sample with two other HGSC Patient 4 and 5 samples, which had readily available PAX8- cells (see https://ianevskialeksandr.github.io/figovfig145.png). The integration was achieved using the standard Seurat workflow, and the cell types were assigned using ScType. Both combined PAX8 + and PAX8- cell populations were visualized using Seurat "FeaturePlots". We used an average of previously-measured responses of PAX8- cells from patient 4 and 5 samples (serving as combined ovarian-sample normal controls) to 372 compounds overlapping with the LINCS 2020 compounds. We extended these initial analyzes with two additional HGSC patients (Patients 2 and 3) for which both organoids and the stromal cell cultures were available at the cell numbers sufficient for single-

drug sensitivity and selectivity testing in PAX8 + and PAX8- cells (Suppl. Table 3).

## Prospective testing in ovarian tumor organoids and drug response assays

In contrast to the AML case, where we had enough cancer cells for combinatorial testing of targeting two major subclones, in the HGSC case study, we opted to predict multi-targeting monotherapies, due to the small proportion of patient-derived cancer cells based on the scRNA-seq analyzes (Fig. 3a–c). In such limited cancer cell populations, identifying subclones may not be reliable for combinatorial targeting. Instead, we used all cancer cells from a patient sample as a collective entity, disregarding subclone distinctions and subdivisions, when identifying treatments that inhibit mostly PAX8+ cells. To predict the compounds that specifically target and eliminate cancer PAX8+ cells, while sparing PAX8- cells, we utilized the differentially expressed genes (DEGs) from the comparison between PAX8+ and PAX8- cells in the scRNA-seq data. These DEGs were used as input for the pre-trained LightGBM model. Among the predicted 372 compound responses (that overlapped with drugs tested on PAX8- cells), we selected the top-20 most effective compounds, and removed two with low confidence, hence resulting in 18 predicted agents. Subsequently, we validated the efficacy of these compounds in PAX8+ tumor organoids and compared the results, as shown in Fig. 3 (3 replicates).

Ovarian cancer organoids were established and characterized according to established protocols[27], and propagated in BME-2 matrix droplets in the sample-specific growth medium. The organoid cultures consisted only of cancer cells as judged by whole-genome sequencing of the organoid cultures, profiling of their copy-number variation, and determination of the tumor cell purity based on the sequencing-based factors in the original study[27]. Moreover, TP53 mutation analyzes, revealing a Variant Allele Frequency (VAF) of 1.0 in all organoids, confirmed that organoids comprise only cancer cells[27].

For the organoid drug sensitivity testing, the organoid cultures were trypsinized to obtain the single-cell suspension. The cells were resuspended in the fresh gel, dispensed to 384-well Ultra-Low Attachment microplates (#4588, Corning) at approximately $10^3$ cells per well in 10 μl of the matrix, and covered with 40 μl of growth medium containing 5 μM ROCK inhibitor to facilitate the organoids formation. After 2–6 days, the medium was exchanged to ROCK inhibitor-free growth medium. Drug testing was performed as described above for single-agent AML sample testing, with the following modifications. The tested compounds (10-fold dilution series of four to five concentrations), vehicle (DMSO), or positive control compounds (100 μM benzethonium chloride or 10 μM bortezomib) were transferred to the wells using Echo 550 acoustic dispenser (Labcyte). The organoids were incubated with drugs for a total of 7 days (with fresh medium exchange and drug replenishment on after 4 days) in the humidified incubator at 37 °C and the viability was assessed using CellTiter-Glo 3D Cell Viability Assay (#G9683, Promega) using a SpectraMax Paradigm microplate reader (Molecular Devices) after 5 min of agitation and 25 min of incubation at room temperature, as indicated by the manufacturer.

The PAX8-negative cells from the ovarian tumor samples were expanded in sample-optimized media, either RPMI-1640 medium, supplemented with 2 mM glutamine, 1% Pen/Strep and 10% FBS (Gibco), or M199 supplemented with 10% FBS, 1% Pen/Strep, 10 ng/mL EGF, 400 nM hydrocortisone, 870 nM insulin-transferrin-selenium, 0.3% Trace elements B, and 20 mM HEPES. Drug testing was performed as above. The culture was trypsinized, resuspended in fresh medium and seeded at 1000 cells in 25 μl of medium per well in pre-drugged 384-well microplates (#3864, Corning). After 7 days of the drug treatment, the viability was measured using the CellTiter-Glo 2.0 (Promega).

## Immunofluorescence and immunoblotting in ovarian cancer organoids and cells

Cells were fixed with 4% PFA for 30 min, washed 5 times with PBS, and incubated in the blocking buffer (PBS, 0.5% BSA, 20 mM glycine, 0.1% TX-100) for 12 h (for organoids) or 1 h (for cell lines) at room temperature. Immunostaining with anti-PAX8 rabbit polyclonal antibody (dilution 1:100, Proteintech, #10336-1-AP) was performed overnight at 4 °C. After 3 washes with the blocking buffer, 8 h each, the donkey-anti-rabbit Alexa555 secondary conjugates (dilution 1:500, Invitrogen, #A31572) were applied for 1 h together with Hoechst 33342 (10 μg/mL in PBS). The cells were imaged at an Opera Phenix confocal screening microscope (Perkin Elmer), with the 20x or 40x water immersion objectives.

For immunoblotting for PAX8 and EpCam, the cells were lysed in RIPA buffer with Pierce protease and phosphatase inhibitors cocktail (Thermofisher) on ice for 30 min. After centrifugation (17000 rcf, 4 °C, 20 min), the samples were loaded to Bis-Tris 4–12% gradient Bolt PAGs and run according to the manufacturer's manual. After the protein transfer to the nitrocellulose membranes overnight in Towbin transfer buffer, the membranes were stained blocked in 5% non-fat milk in TBS-T and incubated with primary antibodies in 5% non-fat milk in TBS-T overnight at 4 °C. The antibodies were: anti-PAX8 rabbit polyclonal (dilution 1:250, Proteintech, #10336-1-AP); anti-EpCam mouse monoclonal (dilution 1:500, Santa Cruz, #sc-25308); and anti-GAPDH mouse monoclonal (dilution 1:5000, Novus, #NB300-221). After 3 washes with TBS-T, the membranes were incubated with the secondary fluorophore-conjugated antibodies diluted to 1:5000 in 5% non-fat milk in TBS-T (anti-mouse IRDye 680, #926-32220; anti-rabbit IRDye 800CW, #926-32211; anti-mouse IRDye 800CW, #926-32210, Licor), washed in TBS-T 3 times and scanned using LiCOR Odyssey imager.

## Statistics & Reproducibility

The statistical tests applied and the significance values are included in the figure legends or results text. We used non-parametric tests in all the statistical comparisons. Patients were recruited as part of two ongoing translational studies for AML and HGSC. The patient samples for the current study were selected based on the availability of input data for modeling (scRNA-seq) and primary cells for experimental validations (single-cell drug assays). The model predictions and experimental testing were done for each patient separately, using scRNA-seq data and single-cell drug assays, respectively. Therefore, the age, sex, gender, race, ethnicity, or other social parameters are not considered as confounding factors. The patient characteristics are reported in Suppl. Tables 1–3. The experimental validations of the model predictions were made after the predictions, and experimental researchers were blinded to the model prediction outcomes. The validation experiments were not randomized, since this is not a case-control study, instead the non-cancerous cells from each patient sample were used as patient-specific control for the particular patient's cancer cell responses. No data were excluded from the analyzes. All the validation drug assays were replicated either 2 or 3 times, depending on the availability of primary patient cells. All the replicate measurements were successful in the sense that the standard deviations were within an expected range based on previous studies.

## Single-cell RNA sequencing and data processing

Single-cell gene expression profiles were generated using the 10x Genomics Chromium Single Cell 3' RNA-seq platform with Next GEM v3.1 Dual Index chemistry. Libraries were prepared using the Chromium Next GEM Single Cell 3' Gene Expression version 3.1 Dual Index kit. Samples were sequenced on an Illumina NovaSeq 6000 system with read lengths of 28 bp (Read 1), 10 bp (i7 Index), 10 bp (i5 Index), and 90 bp (Read 2). Data processing and analysis were performed using 10x Genomics Cell Ranger v6.0.0 pipelines[69]. The "cellranger mkfastq" command, utilizing Illumina's bcl2fastq v2.2.0, was used to generate FASTQ files from raw base calls. The "cellranger count" pipeline performed alignment against the human genome GRCh38.

## Reporting summary

Further information on research design is available in the Nature Portfolio Reporting Summary linked to this article.

## Data availability

The previously published single-cell RNA sequencing data for 9 AML patients are available in the European Genome-Phenome Archive (EGA) under accession codes: EGAS00001004614[33] (AML patients 2, 3, 8 and 10) and EGAS00001004444[70] (AML patients 1, 4, 7, 8, and 11). The single-cell RNA sequencing data generated in this study for AML patients 5, 6, and 12 are available through Sequence Read Archive (SRA accession numbers; Patient 5: SRR30720408; Patient 6: SRR30720407:Patient 12: SRR30720406). The processed Seurat objects were deposited to Zenodo (https://doi.org/10.5281/zenodo.13340927)[71]. The previously published single-cell RNA sequencing data for the 5 HGSC patient samples are available on EGA under accession codes: EGAS00001005010[25] (HGSC patients 1, 2, 3 and 4) and EGAS00001005066[26] (HGSC patient 5). The publicly available data used in this study are accessible in the Connectivity Map LINCS 2020 (https://clue.io/data)[22], PharmacoDB (https://pharmacodb.ca)[23], and PubChem Compound database (https://www.ncbi.nlm.nih.gov/pccompound)[53]. The scRNA-seq data from cancer patients in other tumor types were obtained from the dataset curated by Gavish et al. (https://www.weizmann.ac.il/sites/3CA)[28]. The source data generated in this study are provided in the Supplementary Information or Source Data file. The remaining data are available within the Article, Supplementary Information or Source Data file. Source data are provided with this paper.

## Code availability

The R codes for reproducing the results and for making new patient-specific predictions in other studies are freely available both on GitHub (https://github.com/kris-nader/scTherapy) and Zenodo (https://doi.org/10.5281/zenodo.13340796)[72]. Docker image that encapsulates all the relevant dependencies and ensures compatibility across different environments is available on Docker Hub (https://hub.docker.com/r/kmnader/sctherapy). Separate docker image compatible with the latest Seurat version v5 has also been available (https://hub.docker.com/r/kmnader/sctherapy_v5).

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

## Acknowledgements

We are most grateful to the patients and their families for participating in the studies. The drug assays were carried out at the FIMM High Throughput Biomedicine Unit, hosted by the University of Helsinki and supported by HiLIFE and Biocenter Finland, as well as at the High Content CRISPR screens core facility at BRIC, University of Copenhagen and that is supported by an infrastructure grant from the Novo Nordisk Foundation (grant NNF20OC0061734). We thank the Finnish Hematology Registry and Clinical Biobank (FHRB) for providing the AML biospecimens and associated patient data. Access to HGSC tumor material and data was kindly provided through a collaboration with Prof. Sampsa Hautaniemi, University of Helsinki, Dr. Johanna Hynninen, Turku University Hospital, and the European Union's Horizon 2020 research and innovation program project DECIDER (grant agreement no. 965193). Single-cell RNA sequencing of the patient samples was performed at FIMM Single-Cell Analytics unit supported by HiLIFE and Biocenter Finland. The authors thank Minna Suvela for preparation of the AML patient samples for scRNA-seq, Dr. Juho Miettinen for help with the AML single-cell data, Dr. Imre Västrik for help with the AML patient data, Laura Gall-Mas for help with the HGSC organoid cultures, Dr. Matias M. Falco and Yingjia Chen for their help with the HGSC single-cell data, and Hanna Nebelung for beta-testing the scTherapy codes and docker images. Funding sources: KN: Funding from the Nordic EMBL Partnership Hub for Molecular Medicine, NordForsk (grant #96782). AKG: The Foundation for the Finnish Cancer Institute, Otto A Malm Foundation, Blood disease Research Foundation, K. Albin Johanssons stiftelse sr Foundation, Maud Kuistila Memorial Foundation. AV: Academy of Finland project No 351196 and ERA PerMed JTC2020 project PARIS/Academy of Finland project No. 344697; the Cancer Society of Finland, and the Sigrid Jusélius Foundation. MK: The Foundation for the Finnish Cancer Institute, Cancer Foundation Finland, Finnish Medical Foundation. CH: Research Council of Finland (grants 334781, 320185, 352265 and 357686), Cancer Foundation Finland, Sigrid Jusélius Foundation, and Novartis. TR: Government Research Funding/Helsinki University Hospital. PS: Funding from University of Helsinki Doctoral program, Instrumentariumin tiedesäätiö. TA: European Union's Horizon Europe Research & Innovation program (REMEDi4ALL project, grant agreement No 101057442), European Union's Horizon 2020 Research and Innovation Program (ERA PerMed JAKSTAT-TARGET and CLL-CLUE projects), Research Council of Finland (grants 326238, 340141, 344698, and 345803), Novo Nordisk Foundation Interdisciplinary Synergy Program 2021 (grant NNF21OC0070381), Norwegian Health Authority South-East (grants 2020026 and 2023105), the Cancer Society of Finland, the Norwegian Cancer Society (grants 216104 and 273810), the Sigrid Jusélius Foundation, and iCAN—Digital Precision Cancer Medicine Flagship (iCAN-MULTIDRUG). KW: Novo Nordisk Foundation (grant no. NNF21OC0070381), ERA PerMed JTC2020 project PARIS/Innovation Fund Denmark (grant no. 0204-00005 A). LM-G: Marie Skłodowska-Curie Actions Postdoctoral Fellowship (HORIZON-MSCA-2021-PF-01, 101063359). EP: Research Council of Finland (grant 322675). WS: European Union's Horizon 2020 research and innovation program (grant agreement no. 845045 for RESIST3D), Danish Cancer Society (grant no. R204-A12322). KD: European Union's Horizon 2020 research and innovation program under the Marie Sklodowska-Curie (grant agreement No. 101034291 for DISCOVER).

## Author contributions

A.I., K.N.: conceived the approach; developed the methodology; implemented and tested the method; analyzed the experimental data; prepared the figures; and drafted the manuscript. K.D., W.S., D.B., L.M.-G.: performed the experimental studies on ovarian cancer organoid models and wrote the methods. T.R., H.K., N.I., P.S.: performed the experimental studies on AML patient samples and wrote the methods. M.V.-K., A.K.G.: contributed to the single-cell data of AML patient samples. A.V.: contributed to the single-cell data of ovarian cancer patient samples. M.K., K.P.: contributed to the AML patient samples and clinical data. E.P.: contributed to the clonal analyzes. C.A.H.: contributed to the single-cell data and AML patient samples; commented on the manuscript. K.W.: contributed to the study design, analysis of the results, and writing of the manuscript; supervised the experiments on ovarian cancer organoids. T.A.: supervised the study and wrote the manuscript. All authors approved the final version of the manuscript.

## Competing interests

KP: Research funding from BMS/Celgene, Incyte, Pfizer, and Novartis, unrelated to this study. CAH: Research funding from Kronos Bio, Novartis, Oncopeptides, WNTResearch, and Zentalis Pharmaceuticals for work unrelated to this study, honoraria from Amgen, and personal fees from Autolus. MK reports personal fees from Astellas Pharma, AbbVie, Bristol-Myers Squibb, Faron, Jazz Pharmaceuticals, Novartis and Pfizer and research funding from AbbVie outside the submitted work. TA: Research funding from the European Union's Horizon Europe Research & Innovation program under grant agreement No 101057442. Views and opinions expressed in this document are those of the authors only. They do not necessarily reflect those of the European Union, which cannot be held responsible for the information it contains. All the other authors declare no potential competing interests.

## Additional information

[1]Institute for Molecular Medicine Finland (FIMM), HiLIFE, University of Helsinki, Helsinki, Finland. [2]Biotech Research and Innovation Centre (BRIC), University of Copenhagen, Copenhagen, Denmark. [3]Department of Hematology, Helsinki University Hospital Comprehensive Cancer Center, Helsinki, Finland. [4]Foundation for the Finnish Cancer Institute (FCI), Helsinki, Finland. [5]Research Program in Systems Oncology, Research Programs Unit, Faculty of Medicine, University of Helsinki, Helsinki, Finland. [6]iCAN Digital Precision Cancer Medicine Flagship, University of Helsinki and Helsinki University Hospital, Helsinki, Finland. [7]Applied Tumor Genomics Research Program, Research Programs Unit, Faculty of Medicine, University of Helsinki, Helsinki, Finland. [8]Institute for Cancer Research, Department of Cancer Genetics, Oslo University Hospital, Oslo, Norway. [9]Oslo Centre for Biostatistics and Epidemiology (OCBE), Faculty of Medicine, University of Oslo, Oslo, Norway. [10]These authors contributed equally: Aleksandr Ianevski, Kristen Nader.
✉e-mail: tero.aittokallio@helsinki.fi

