## [Peer Review File · Nature Communications]

Single-cell transcriptomes identify patient-tailored therapies for selective co-inhibition of cancer clonesREVIEWER COMMENTS

Reviewer #1, expertise in sc-RNAseq, machine learning and drug response prediction
(Remarks to the Author):

The authors propose scTherapy, a machine learning approach that dissects intratumoral heterogeneity with single cell RNA-Seq, identifies genetically distinct malignant cell populations and proposes personalized drug combinations tailored towards each tumor clone. scTherapy has been trained on small molecule perturbation assays (LINCS 2020) and viability response data mined from PharmacoDB. Additionally, the model takes into account the molecular structure of the compounds and the doses at which they have been tested.

Ianevski and colleagues have validated their model in single cell RNA-Seq samples from AML patients and in ovarian derived tumor organoids, suggesting that scTherapy drug combinations exhibit low toxicity in normal (healthy) cells, and synergistic effects against malignant populations. In this regard, the authors should be commended for including a detailed clinical history of the AML patients involved in the experimental validation of scTherapy.

In general the manuscript is well written, fluent, clear and understandable. In my opinion the manuscript is an elegant example of the application of in silico drug prescription methods in the single-cell transcriptomics scenario and tackles for the first time the challenges of dosage and toxicity in the context of drug combinations. However, I have some major concerns about the background in the area described by the authors, the reproducibility and usability of scTherapy as well as some of the results showcased in the manuscript.

Major

The authors should expand the computational work adding more results in other publicly available single-cell data to show versatility and scope of the method in other tumour types. Including those that already have treatment data, primary vs metastasis samples or samples with some additional clinical context. Actually, there are several resources with pan-cancer single cell data that can be used to test scTherapy and evaluate the results (Gavish et al. Nature 2023). It would be very interesting and useful to know if there are recurrent combinations between patients from the same tumour type or if personalized medicine is the most frequent option.

The authors should explain why they have defined subclones based on genetics (i.e.CNV) and not using expression clusters only - what difference would it make? Have they done any comparison in this sense?

It is not clear to me how drugs are prioritized and how specific and/or global is that prioritization taking into account the different subclones: How many clones are targeted with the proposed drugs per patient? What is the scope of the proposed drugs? Do they target the majority clone or minor populations? Which combination(s) is the most effective

in targeting the largest number of clones?

A more thorough introduction of the state of the art on single cell and cancer is needed to improve the manuscript. TME is not properly addressed as well.

In the introduction, a more thorough introduction of the state of the art on single cell and cancer is needed to improve the manuscript. TME is not properly addressed as well. Additionally, the authors make a claim stating that "Several computational approaches for in silico prediction of drug combination effects have been developed, yet we lack approaches that consider both the patient and disease heterogeneity when predicting drug sensitivity differences among cell populations, toward designing cancer selective and patient specific therapeutic options using feasible measurements in scarce patient cells". At this point I miss some references to previous valuable methods and efforts published in the area using single-cell transcriptomics (e.g. PMID: 35105355; PMID: 34911571; PMID: 36928331; PMID: 36833194; PMID: 36544472).

The authors should compare their results for monotherapies with those provided by other in silico prescription methods previously reported.

The authors should provide some kind of containerization for scTherapy. In its current state, the tool depends not only on common R libraries but also depends on packages which must be downloaded directly from GitHub. Moreover, it is not clear to me which specific versions of each library should be installed. In these cases, conflicts arising from incompatible package versions are common and seriously hamper reproducibility of the results showcased in the paper and the usability of the proposed methodology. Including a docker image or even a conda recipe would help potential users to install and use scTherapy. I was unable to install the required dependencies following the exact instructions laid out in the GitHub repository, most probably because of yaGST requiring an older and incompatible R version compared to my current setup.

Regarding the experimental validation in AML patients, Supl. Table 1. suggests that relapsed, refractory and newly diagnosed patients were considered for the analysis. I wonder about the performance of scTherapy when comparing the newly diagnosed and otherwise treatment naive patient with the relapsed and the refractory patient. Have the authors found any differences regarding the amount of available candidates and the dosages required to reach malignant inhibition in newly diagnosed against refractory patients? Does scTherapy identify a reduction in malignant inhibition for anagrelide in patient 12? In summary, is scTherapy capable of capturing the therapeutic heterogeneity arising from previous treatments? Is the standard of care offered by scTherapy? Does it usually come out? To what extent the results found would improve the standard of care?

Interestingly, the authors find that there are different clonal subpopulations that share potential treatments. This has been reported in previous articles (see the concept of "therapeutic clusters" introduced by Fustero-Torre et al. (PMID: 34911571)). The Discussion section could be enriched by discussing this point along with the contributions of scTherapy.

Consider shortening and structuring the "Discussion" section, as it is quite lengthy (almost

resembling the Results section) and sometimes lacks substantial content.

Minor

In the methods section, the authors state that 10303 overlapping compound-cell lines were considered for training scTherapy. However, it is not clear to me whether all of these drugs are then considered as potential candidates in the tool. That is, which is the universe of compounds that scTherapy is able to propose? A brief statement or a small addition to Fig 1. would be highly informative in this regard.

In the discussion, the authors claim to have evaluated already existing machine learning approaches, but do not elaborate on specific tools nor publications, citing instead a review. Consider adding a brief statement about competing methods to improve the discussion and to highlight how scTherapy fits.

The authors claim that their approach is a widely-applicable strategy to prioritize personalized therapeutic candidates. As far as I understood, scTherapy relies mostly on the gene expression profile of each malignant population to find compounds able to reverse the disease state. However, I wonder about tumor types for which the most predictive feature for a given targeted therapy is actually a point mutation. For instance, melanomas harboring the BRAFV600E mutation benefit from BRAF inhibitors such as vemurafenib. However, studies of the predictive power of gene expression in this context have showcased that gene expression has little relative importance for predicting drug response, being the presence of the mutation the most useful feature. See: <https://depmap.org/portal/compound/VEMURAFENIB?tab=predictability>.

Reviewer #2, expertise in ovarian cancer organoids (Remarks to the Author):

This is an interesting and ambitious study for the development of new tools for improving data-driven decision-making approaches in oncology. The authors developed a new computational model (Light GBM) and trained it on available data sets containing data on genome-wide transcriptomic profiles from 167 cell lines (LINCS2020) project and added data on viability responses from the Pharmacology DB platform.

The experimental part of the study is mainly focused on testing predicted treatments in the AML patients group. The new sequencing data set from 12 bone marrow samples has been produced and, comprehensive in vitro validation performed. In line with this and supporting the basic premise of the study to select optimal multitarget therapy to address cancer heterogeneity and defined personalized treatment in each case authors identify combined top effective and selective compounds for each patient. In vitro data includes a comparison of single effective single agents vs. noneffective agents, and also tests of combinatorial two-drug treatments for 4 patients. The figures Figures 2b.d.e. convincingly demonstrate the ability of the model to pick effective drug combinations for AML treatment

In the case of ovarian cancer experimental data is limited and significantly less convincing.

While the wish of authors to extend the application field of their tool to solid malignancies is understandable, the presented data has significant weaknesses and limitations, to support the overall conclusions of the study.

Some of the major issues are:

- The validation is performed only with one organoid line from a single patient. This conflicts with the main claim of the study to select a treatment for successful targeting of cancer heterogeneity. To achieve that one would have to provide evidence by testing predicted combinations on several clonal lines from a single patient included in the experimental setup.
- Independently of heterogeneity issue, single patient validation strategy is inadequate.
- In the case of ovarian cancer the authors decided to perform only a comparison of a single treatment between cancer and noncancer cells (fibroblast) citing limitations in the cell number of organoids. This is explanation is not conceivable. PDO organoids, if properly cultivated, are robust models and should be easily expandable to a multiwell format and multiple plates which would easily allow multi-drug combinations testing. The source publication authors cite and where they obtained organoids from does show stable long-term growth so it is not understandable why there should be any limitations.
- Also, it is not clear why a comparison of effective vs. ineffective treatment, similar to Fig 2b is omitted in the case of the ovarian cancer model. This is should be straightforward and doable.
- The usage of fibroblasts as “normal cells” is not ideal as it essentially involves a comparison of 2D vs. 3D cell culture models which is likely a confounding factor. It would be much better if the authors would make a comparison to healthy organoids from the same patient when ovarian cancer is taken. I understand that this point would be difficult to address without obtaining fresh samples, nevertheless, the issue needs to be addressed or at least in-depth discussed beyond brief mention in the current version of the manuscript.
- The description of the processing of sc RNA data of ovarian cancer correctly postulates that PAX8 expression can be used as a discriminatory marker to distinguish cancer vs. noncancer cells in the tissue as PAX8 is uniformly expressed in cancer tissue. Organoids represent only the epithelial compartment and thus should be strongly PAX8 positive. Authors state that organoids contain only cancer cells. Still, Figure 3c of treatment naïve tumor organoids shows an image that contains multiple cells that are PAX8 negative. What is the origin of these cells? Are all cells Epcam positive?

Minor point

- The abstract is very general and should be amended to be more specific and closely related to the experimental data.

Overall this is an interesting study potentially of interest for the broader research and medical community interested in drug treatment predictions and the development of personalized therapies. However, the prediction model requires more robust wet lab testing.

In particular for its application for the treatment of solid malignancies, in this case, ovarian

cancer. Expanding the number of tested organoid lines beyond $n=1$ and testing of combinatorial treatments as well as comparison of effective vs non effective drugs should be prioritized to adequate experimental validation data to support the overall conclusions of the study.

Reviewer #3, expertise in clonal evolution and bioinformatics (Remarks to the Author):

Advanced malignancies often exhibit complex intratumor cellular diversity and require therapies that target multiple aspects of the disease. However, the high number of potential drug/dose combinations makes identifying patient specific treatments challenging. In this work, the authors develop a machine learning approach which exploits single-cell transcriptomic data to prioritize personalized multi-targeting treatments for patients with hematological cancers or solid tumors. The proposed framework was applied to a dataset comprising single-cell RNA-seq data for 12 AML patients and to single-cell data for 3 ovarian cancers and identified patient-specific treatment combinations targeting various biological pathways. Some of the results were experimentally validated.

Overall, I believe personalized treatment to be a crucial aspect of cancer care, making this a significant topic. However, I have a few concerns I would like the authors to address.

1) The manuscript is generally well-written. However, I recommend some improvements to the structure, particularly within the Introduction, Results, and Discussion sections. In the current draft, the initial portion of the Introduction appears to be primarily focused on hematological malignancies, which is fine. However, later in the paper, the authors discuss the significance of their approach in the context of practical limitations for a broad drug testing on solid tumors. In support of this, the authors present a case study on ovarian cancer.

To enhance the manuscript's clarity and relevance, I suggest expanding its framing to encompass a broader context. Specifically, it would be beneficial to provide a more comprehensive introduction that highlights the broader implications of the study, not only for hematological malignancies but also for solid tumors. Emphasizing the significance of their approach across various cancer types, with a clear rationale, will help readers better appreciate the relevance and potential impact of the research.

2) In my opinion, the Discussion section is quite long and contains introductory concepts, such as the limitations of the current state-of-the-art machine learning approaches for predicting anticancer drug combinations. I believe it would be more effective to present these introductory notions in the Introduction section instead.

I would recommend that the authors perform a reorganization of the manuscript to provide a more comprehensive introduction to the problem. This should encompass both the biological/wet lab aspects of the topic and the bioinformatic approaches/literature they intend to improve and why.

Moreover, I would ask the authors to better discuss the novelty of ScTherapy with respect to previously published approaches.

3) The ScTherapy framework includes a series of preprocessing steps, starting with raw scRNA-seq counts derived from samples obtained from cancer patients. These preprocessing steps are designed with two primary objectives: to establish distinct clusters of cells, distinguishing between normal cells and clones/subclones within the dataset. And, to identify and compile a list of differentially expressed genes that characterize the specific clones with respect to the normal cells.

I have two concerns regarding this part, which I believe the authors should carefully address in their manuscript. First, it would be valuable to understand the impact of the stability of the detection process differentiating between normal cells and the various clones. Given that ScTherapy relies on differentially expressed genes to characterize each clone relative to normal cells for drug prediction, it's crucial to assess how uncertainties in clone detection might impact the method's reliability. A discussion on the robustness of this process and potential strategies to mitigate uncertainties would enhance the manuscript's rigor.

Second, considering the objective of personalized medicine and the substantial heterogeneity in cancer, the authors should provide insights into how ScTherapy could be adopted to offer treatment suggestions for new patients in relation to the detected clones for which candidate therapies are designed. Exploring methods to generalize ScTherapy predictions beyond the studied cases and addressing the challenges of extending personalized treatment recommendations to diverse patient populations would strengthen the manuscript's applicability and relevance in clinical contexts. Should the absence of genetic information be considered a limitation? I recommend that the authors discuss this topic in their manuscript.

4) I appreciate the authors' effort in constructing a drug-response database and in employing advanced machine learning techniques to address the drug prediction task. Nevertheless, in the current manuscript, there is no quantitative assessment regarding the performance of their method. While I understand that this might not be straightforward, I believe it's essential to include quantitative data, possibly through simulations, to validate the effectiveness of both the method and its current implementation.

5) As a final comment, I would like to discuss the authors' claim regarding the particular significance of their approach in the context of drug prediction for solid tumors. I do not argue against this statement. However, I would like to notice that the current experimental results presented in the manuscript are predominantly emphasized on hematological cancers. While I understand that conducting experiments on solid tumors, such as ovarian cancer, might be resource intensive, however I believe that a more extensive set of experiments in this context could significantly enhance the manuscript's value and significance.

Reviewer #4, expertise in AML functional genomics and clonal evolution (Remarks to the

Author):

In this paper, the authors use scRNA-seq and machine learning to identify combinatorial therapies that could potentially target multiple clones in a single patient's cancer. This (semi)automated approach would have excellent potential for clinical delivery, although significantly more validation would be required. Nevertheless, the approach is innovative and goes in the direction that is needed to ultimately provide curative therapies for cancer patients.

This is a very well written paper addressing an important problem.

My main issue is that in Figs 2e and 3d, there is no statistical comparison to show whether the differences between normal and malignant cells are significant or not. Given that there are replicates (although details of this are unclear), the authors should derive p-values and then revise the numbers estimating the proportions of samples that identify combination therapies that could be efficacious as well as the proportions that would be differentially targeted compared to normal cells.

The other issue is that while it makes sense the drugs and dosages provided are in reference to the malignant cells, this is not necessarily what would be reflected in the clinic. The authors should discuss the point that dose adjustments are often required depending on age, comorbidities and other issues.

REVIEWER COMMENTS

Reviewer #1, expertise in sc-RNAseq, machine learning and drug response prediction (Remarks to the Author):

The authors propose scTherapy, a machine learning approach that dissects intratumoral heterogeneity with single cell RNA-Seq, identifies genetically distinct malignant cell populations and proposes personalized drug combinations tailored towards each tumor clone. scTherapy has been trained on small molecule perturbation assays (LINCS 2020) and viability response data mined from PharmacoDB. Additionally, the model takes into account the molecular structure of the compounds and the doses at which they have been tested.

lanevski and colleagues have validated their model in single cell RNA-Seq samples from AML patients and in ovarian derived tumor organoids, suggesting that scTherapy drug combinations exhibit low toxicity in normal (healthy) cells, and synergistic effects against malignant populations. In this regard, the authors should be commended for including a detailed clinical history of the AML patients involved in the experimental validation of scTherapy.

In general the manuscript is well written, fluent, clear and understandable. In my opinion the manuscript is an elegant example of the application of in silico drug prescription methods in the single-cell transcriptomics scenario and tackles for the first time the challenges of dosage and toxicity in the context of drug combinations. However, I have some major concerns about the background in the area described by the authors, the reproducibility and usability of scTherapy as well as some of the results showcased in the manuscript.

Our response: We thank the reviewer for appreciating our work, and its novelty for in silico drug prescription. We have addressed all the comments and provide our point-by-point responses below. All the text changes in the revised manuscript are highlighted in yellow.

Major

The authors should expand the computational work adding more results in other publicly available single-cell data to show versatility and scope of the method in other tumour types. Including those that already have treatment data, primary vs metastasis samples or samples with some additional clinical context. Actually, there are several resources with pan-cancer single cell data that can be used to test scTherapy and evaluate the results (Gavish et al. Nature 2023). It would be very interesting and useful to know if there are recurrent combinations between patients from the same tumour type or if personalized medicine is the most frequent option.

Our response: We appreciate the reviewer's suggestion to show scTherapy's versatility with different tumor types. We acknowledge the resource by Gavish et al and agree that using single cell datasets from various tumors can offer insights on recurring therapy options within the same tumor type and the prevalence of predicted personalized treatments. As a result, we have

expanded our analysis using the Gavish et al dataset and applied scTherapy to 24 additional patients from 3 tumor types spanning various disease and treatment stages; 10 patients with lung adenocarcinoma (LUAD, 5 primary samples, 5 metastatic, 4 treatment naïve, 6 treated), 10 patients with pancreatic ductal adenocarcinoma (PDAC, 5 primary samples, 5 metastatic), and 4 patients with triple negative breast cancer (TNBC, 4 treatment naïve primary samples):

1. Laughney, A. M., Hu, J., Campbell, N. R., Bakhoun, S. F., Setty, M., Lavallée, V.-P., Xie, Y., Masilionis, I., Carr, A. J., Kottapalli, S., Allaj, V., Mattar, M., Rekhman, N., Xavier, J. B., Mazutis, L., Poirier, J. T., Rudin, C. M., Pe'er, D., & Massagué, J. (2020). Regenerative lineages and immune-mediated pruning in lung cancer metastasis. *Nature Medicine*, 26(2), Article 2. <https://doi.org/10.1038/s41591-019-0750-6>
2. Lin, W., Noel, P., Borazanci, E. H., Lee, J., Amini, A., Han, I. W., Heo, J. S., Jameson, G. S., Fraser, C., Steinbach, M., Woo, Y., Fong, Y., Cridebring, D., Von Hoff, D. D., Park, J. O., & Han, H. (2020). Single-cell transcriptome analysis of tumor and stromal compartments of pancreatic ductal adenocarcinoma primary tumors and metastatic lesions. *Genome Medicine*, 12(1), 80. <https://doi.org/10.1186/s13073-020-00776-9>
3. Gao, R., Bai, S., Henderson, Y. C., Schalck, A., Yan, Y., Kumar, T., Hu, M., Sei, E., Davis, A., Wang, F., Shaitelman, S. F., Wang, J. R., Chen, K., Moulder, S., Lai, S. Y., & Navin, N. E. (2021). Delineating copy number and clonal substructure in human tumors from single-cell transcriptomes. *Nature Biotechnology*, 39(5), 599–608. <https://doi.org/10.1038/s41587-020-00795-2>

These cancer types were chosen as they pose clinically significant therapeutic challenges; in particular, patients with PDAC, LUAD and TNBC urgently need new targeted treatment options. As for reference, we also included 12 samples from our AML cohort reported in this manuscript (4 with experimental combination response validations), as well as 4 ovarian cancer samples from our HGSC cohort of metastatic tumors with poor responsiveness to standard chemotherapy, representing a highly challenging case for personalized treatment identification.

Our analysis, depicted in the new **Figure 4** (see also below), demonstrates the presence of therapy clusters shared among patients within the same tumor type, as well as emergence of unique, patient-specific therapies. These findings provide a statistical landscape of the predicted drugs: 19% are patient-specific, indicating a high degree of personalized treatment potential, 25% are disease-specific (2% LUAD, 1% TNBC, 2% PDAC, 10% AML, 10% HGSC), and 22% were common across all cancers. This distribution underscores the versatility of scTherapy in addressing the diverse and complex landscape of cancer treatments, paving the way for more targeted and effective therapeutic strategies across a range of tumor types.

In summary, our expanded analysis demonstrates a balance between recurrent treatment options within tumor types and a strong inclination towards personalized medicine, with 19% of treatments being patient-specific. Additionally, we observed that 22 out of the 131 predicted treatments have gone through either phase 3 or 4 clinical trials (new **Supplementary Table 4**), underscoring the capability of the prediction approach to identify clinically relevant and potentially efficacious treatment options for personalized cancer therapy. We note that many of the scTherapy predictions also involve investigational compounds that could be explored in future clinical studies.

Supplementary Table 4 | Clinical development stages of scTherapy-predicted treatments across 5 cancer types (<http://clinicaltrials.gov/>).

Indication	Treatment	Phase	NCT identifier number
Breast	abemaciclib	3	NCT05952557,NCT03155997
	alpelisib	3	NCT05501886,NCT04251533
	epirubicin	4	NCT01642771,NCT01216111,NCT00630032,NCT00540800, NCT01199432 ,NCT00689156,NCT03498716 NCT04301739,NCT03036488, NCT04136782 ,NCT05862064,NCT02455141,NCT04031703,NCT03876886 NCT04296175,NCT01378533,NCT04335669,NCT06112379,NCT00912444
	5-fluorouracil	3	NCT00121992,NCT04031703
Lung	etoposide	3	NCT00003364,NCT00717938,NCT00632853,NCT00002858,NCT00003696,NCT00003606,NCT00003299,NCT00061919,NCT00011921,NCT00433498,NCT00045162,NCT02875457,NCT00002822,NCT00812266
	paclitaxel	3	NCT00003696,NCT00003299,NCT00011921,NCT00003317,NCT00003589,NCT02477826,NCT00054184,NCT00795340,NCT00054197,NCT00006049,NCT00054210,NCT00551733
	epirubicin	3	NCT00003606,NCT00011921
	docetaxel	3	NCT00074204,NCT00022022,NCT00883675,NCT00054184,NCT02076477
	tivantinib	3	NCT01244191
AML	daunorubicin	3	NCT00428558,NCT00589082,NCT04174612,NCT00266136,NCT00931138,NCT00715637,NCT03897127,NCT02013648,NCT00703820,NCT04293562,NCT00927498,NCT00363025
	decitabine	3	NCT03941964,NCT05177731,NCT02172872,NCT05586074,NCT02348489,NCT01633099,NCT02785900
	etoposide	3	NCT00052299,NCT02421939,NCT03504410,NCT03182244,NCT00703820,NCT04293562
	mitoxantrone	4	NCT01828489,NCT00052299,NCT02421939,NCT03504410,NCT03182244,NCT02461537,NCT04293562, NCT

			00180102
	panobinostat	3	NCT04326764
	tipifarnib	3	NCT00093990
HGSC	alpelisib	3	NCT04729387
	cyclophosphamide	3	NCT00003214,NCT00004921,NCT00002477,NCT04520074,NCT00002819,NCT00068601
	etoposide	3	NCT04000295,NCT04520074
	olaparib	4	NCT03737643,NCT02392676, NCT02476968 ,NCT03402841,NCT05255471,NCT03106987,NCT01874353,NCT04729387,NCT03534453,NCT01844986,NCT02282020,NCT04884360,NCT04330040,NCT02477644,NCT03740165
	paclitaxel	4	NCT03737643,NCT05371301,NCT00326456,NCT04729608,NCT00657878,NCT00660842,NCT00003214,NCT03940196,NCT00004921,NCT04000295,NCT03398655,NCT04337632,NCT02718417,NCT01239732,NCT01802749, NCT01706120 ,NCT04729387,NCT00002894,NCT00003644,NCT01684878,NCT03690739,NCT05145218,NCT00002717,NCT06072781,NCT00002819,NCT05009082,NCT00189553,NCT00003322,NCT00003998,NCT03806049,NCT01654146,NCT02470585,NCT05281471,NCT03740165,NCT03794778,NCT00006454,NCT00028743,NCT00483782,NCT02631876,NCT00002568,NCT05601700,NCT04908787,NCT00189371
Pancreas	5-fluorouracil	3	NCT00417209,NCT00602745,NCT05314998
	paclitaxel	4	NCT03721744, NCT04217096 ,NCT01836432,NCT03941093,NCT02506842,NCT04229004,NCT05178628,NCT04617821, NCT05035147 ,NCT05751850,NCT04835064,NCT05653453,NCT04674956,NCT06017284,NCT04329949,NCT04935359,NCT02101021,NCT03943667,NCT02715804, NCT04480268 ,NCT02993731, NCT03401827

Boldfaced NCT IDs represent clinical trials that have reached phase 4.

These new results are now included in the revised manuscript under the new Results section “**Landscape of predicted drug responses in solid tumor and hematological cancers**” (pages 8-10), and in revised Discussion (pages 12). We believe these additional results in pan-cancer single-cell data demonstrate the versatility and scope of the method in multiple tumour types and single-cell datasets, underscoring its potential in advancing personalized cancer therapy.

Fig. 4 | scTherapy predictions across solid tumors and hematological cancers. (a) Heatmap of the scTherapy predicted monotherapies across patient samples from multiple tumor types (columns). Each column represents a patient sample, with annotations indicating treatment history, disease stage, and cancer type. The color coding indicates predicted effectiveness of the treatment (rows). Instances where the ScTherapy did not generate a prediction for a specific patient sample are marked with a grey color. (b) Overlap of predicted treatments between patients with solid tumor (PDAC, TNBC, LUAD, HGSC) and patients with hematological cancer (AML) (c). Overlap of predicted treatments when focusing on the solid tumors only (PDAC, TNBC, LUAD, HGSC).

The authors should explain why they have defined subclones based on genetics (i.e.CNV) and not using expression clusters only - what difference would it make? Have they done any comparison in this sense?

Our response: We thank the reviewer for this valuable question and appreciate the opportunity to clarify our approach. Our focus in this study was to target the different genetic subclones of cancers due to their direct link to the tumor's evolutionary dynamics and their more established role in influencing treatment outcomes and drug resistance, thereby providing a more precise foundation for developing targeted therapies. Hence, we chose to use CNV patterns to define cancer genetic subpopulations based on scRNA-seq-inferred CNV profiles, instead of solely relying on expression-based cell clusters to define genetic subclones, in agreement with recent evidence from other studies showing that transcriptional cell clusters do not necessarily correspond to genetic clones. We agree that the phenotypic subpopulations inferred based on expression clusters may also drive differential drug sensitivities; however, these transcriptional clusters more likely reflect variations in cell types/states, rather than the underlying genetic subclonal diversity within tumors. This distinction is crucial, as our study specifically targets the genetic subclones. This distinction is supported by many recent studies that have observed differences between transcriptional clusters and genetic subclones:

1. Tirier, S.M., Mallm, JP., Steiger, S. et al. Subclone-specific microenvironmental impact and drug response in refractory multiple myeloma revealed by single-cell transcriptomics. *Nat Commun* 12, 6960 (2021). <https://doi.org/10.1038/s41467-021-26951-z>
2. Liu, X., Jin, S., Hu, S. et al. Single-cell transcriptomics links malignant T cells to the tumor immune landscape in cutaneous T cell lymphoma. *Nat Commun* 13, 1158 (2022). <https://doi.org/10.1038/s41467-022-28799-3>
3. Herrera A, Cheng A, Mimitou EP, et al. Multimodal single-cell analysis of cutaneous T-cell lymphoma reveals distinct subclonal tissue-dependent signatures. *Blood* 138, 1456-1464 (2021). <https://doi.org/10.1182/blood.2020009346>

To address the reviewer's question, we conducted a side-by-side analysis in the AML patient sample 12, where we compared subclones identified based on inferred CNVs (upper panel of the figure below, new **Suppl. Fig. 8a**), with those determined based on expression cell clustering (lower panel, **Suppl. Fig. 8b**). Notably, there is a marked decrease in the expression around chromosome 5 in both heatmaps, relative to healthy cells. This suggests significant copy number loss in this region in the AML cells, contributing to the cancerous phenotype. Upon reviewing the patient data, we confirmed that this sample exhibits the del5q karyotype, a recognized recurrent driver alteration in AML, which is associated with a poor prognosis:

1. Volkert S, Kohlmann A, Schnittger S, Kern W, Haferlach T, Haferlach C. Association of the type of 5q loss with complex karyotype, clonal evolution, TP53 mutation status, and prognosis in acute myeloid leukemia and myelodysplastic syndrome. *Genes Chromosomes Cancer*. 2014 May;53(5):402-10. doi: 10.1002/gcc.22151.
2. Jerez A, Gondek LP, Jankowska AM, Makishima H, Przychodzen B, Tiu RV, O'Keefe CL, Mohamedali AM, Batista D, Sekeres MA, McDevitt MA, Mufti GJ, Maciejewski JP. Topography, clinical, and genomic correlates of 5q myeloid malignancies revisited. *J Clin Oncol*. 2012 Apr 20;30(12):1343-9. doi: 10.1200/JCO.2011.36.1824.

When analyzing subclones, the inferCNV-detected subclone E shows notably higher copy numbers around chromosome 19, compared to subclone B, and a similar but inverse difference is observed around chromosomes 7-8 (highlighted in bold areas in the figure). These patterns, distinctly captured in the inferCNV analysis, indicate subclone-specific profiles. In contrast, these distinctions are absent in the standard Seurat cluster analysis, which focuses on individual gene expression profiles, rather than specific CNV patterns. This implies that expression-based clustering misses certain subclonal genomic variations, especially those driven by CNVs, which may be critical for determining clone-specific drug sensitivities. Assessing how much this difference affects treatment response predictions and their experimental validations warrants a separate study, as here we chose to focus on genetic subclonal targeting.

Supplementary Fig. 8 | Genetic clone detection vs. cell clustering. Comparison of cancer subclone characterization using inferred Copy Number Variation (CNV) profiles (upper panel) versus expression-based cell clusters (lower panel) in an AML patient sample 12. Bold rectangles highlight areas with pronounced expression differences, underscoring the effectiveness of CNV-based analysis in uncovering genetic subclonal diversity essential for designing clone-specific treatment options.

We argue that accurate identification of genetic subclones is vital for predicting effective drug treatments for cancer-selective therapies, like was also experimentally validated in the AML and HGSC patient samples. We have now clarified this approach and its benefits in the revised version, but also state that the predictive approach could be used for identifying inhibitors of other phenotypic subpopulations or cell states inferred from the scRNA-seq profiles (pages 14-15):

It is not clear to me how drugs are prioritized and how specific and/or global is that prioritization taking into account the different subclones: How many clones are targeted with the proposed drugs per patient? What is the scope of the proposed drugs? Do they target the majority clone or minor populations? Which combination(s) is the most effective in targeting the largest number of clones?

Our response: We thank the reviewer for these important questions and appreciate the opportunity to clarify our approach, which we agree was not optimally described in the submitted version. Our tool offers flexibility for the users in performing either monotherapy or combination therapy predictions, based on the input scRNA-seq data. In any case, scTherapy does not seek to maximize the number of targeted clones, instead to identify selective targeting options for major cancer clones, while avoiding the inhibition of non-cancerous cells, thereby increasing the likelihood of the predicted multi-targeting treatments for clinical success.

In cases where the proportion of cancer cells in the scRNAseq data is relatively small, identifying subclones may not be meaningful for the experimental validation of combinatorial targeting, given the limited number of cells. For instance, in the ovarian cancer patient case study, we opted to predict multi-targeting monotherapies, instead of combination therapies, due to the small proportion of patient-derived cancer cells (**Fig. 3a-c**). This is because identifying major subclones for combinatorial targeting may not be reliable in those samples. Therefore, we shifted in the HGSC patients our focus to predicting monotherapies over combination therapies, where we used all the cancer cells from a patient sample as a collective malignant entity, disregarding subclone distinctions and subdivisions. This is now explained on page 7 (please see also pages 20-22 of Materials and Methods for the HGSC organoids and experimental assays).

However, when using the tool for predicting combination therapies, like in the AML case study, we shifted our focus to a broader level subclones. Currently, the predictions are limited to two majority clones, where a specific drug in the drug combination is uniquely predicted for each subclone. We acknowledge this confusion and have clarified these details in the revised manuscript (pages 4, 15, 18 and 20). However, as explained below, the current predictions from scTherapy can also target both major and minor subclones, due to polypharmacological effects of the drugs. In future developments, we plan to explore and extend scTherapy for multi-drug combinations to selectively target also other subclones, e.g., three or more subclones, as discussed on page 15. It is important to note that such extensions may lead to higher order combinations of more than two drugs, which poses challenges in experimental validation.

To address the question of drug combination scope and their targeting of major versus minor clones, we have now identified minor subclones in four AML patient samples for which drug combinations were predicted (illustrated in **Fig. 2c**); this new analysis identified 2 major and 4 minor subclones within the cancer cell populations of each sample. We then examined the overlap of drug predictions among these subclones (new **Suppl. Fig. 9**). It was observed that the drug predictions for major subclones largely coincide with those for minor subclones within the same broad-level clone (broad subclone A can be further separated into clones C and D, similarly for broad subclone B into E and F). For example, in Patient 5, a total of 19 drugs were predicted to

target both the broad subclone A and specific subclone D. Thus, the scope of the proposed drugs encompasses both major and minor clonal populations, indicating a relatively comprehensive yet selective targeting strategy across the clonal spectrum of cancer cells, as mentioned on page 15.

Supplementary Fig. 9 | Overlap in scTherapy treatment predictions among the major and minor subclones across four AML patient samples. The darker bars highlight drug prediction overlaps with the first major subclone, and the lighter bars with the another. The colored subclone labels denote the major subclones, while those in gray denoted minor subclones. The four patient samples correspond to those used in the experimental validation.

In the introduction, more thorough introduction of the state of the art on single cell and cancer is needed to improve the manuscript. TME is not properly addressed as well.

Our response: We have now extended the introduction to provide state of the art of single cell profiling in cancer research, and the importance of TME in solid tumors (page 2).

Additionally, the authors make a claim stating that “Several computational approaches for in silico prediction of drug combination effects have been developed, yet we lack approaches that consider both the patient and disease heterogeneity when predicting drug sensitivity differences among cell populations, toward designing cancer selective and patient specific therapeutic options using feasible measurements in scarce patient cells”. At this point I miss some references to previous valuable methods and efforts published in the area using single-cell transcriptomics (e.g. PMID: 35105355; PMID: 34911571; PMID: 36928331; PMID: 36833194; PMID: 36544472).

Our response: We thank the reviewer for pointing out these valuable works, which we have now cited in the revised manuscript (pages 2-3). We note, however, that none of these previous methods enable identification of drugs or drug combinations that co-inhibit genetic clones at a single-cell and individual-patient level. For instance, DEGAS (PMID: 35105355) helps to overlay disease associations onto individual cells, but it does not provide any drug/combinations predictions. Similarly, the study by Zhang et al. (PMID: 36928331) primarily investigates the tumor microenvironment (TME) of hypopharyngeal carcinoma (HPC) at the single-cell level to identify gene modules associated with clinical treatment response and patient prognosis.

Based on a recent review of single-cell level drug response prediction methods [PMID: 38109840], the key current limitations of the existing methods are: (i) they cannot predict drug combination responses, rather focus on single drug effects; (ii) they do not attempt to predict effective drug dosage, which is critical for translational applications; and (iii) they have not been experimentally evaluated to truly evaluate their practical utility. Our work addresses all these limitations of the current methods for single-cell level drug treatment response prediction in cancer applications. This is now mentioned in the revised discussion (page 13).

The authors should compare their results for monotherapies with those provided by other in silico prescription methods previously reported.

Our response: We thank the reviewer for this comment and do acknowledge the importance of benchmarking scTherapy against other methods that have been reported. To address this comment, we have now benchmarked scTherapy monotherapy predictions against two recently implemented methods - BeyondCell (PMID: 34911571) and scDrug (PMID: 36544472) using the 12 AML patient samples described in this manuscript. The experimental data on single-agent responses in these patient samples provides a systematic and fair evaluation of the efficacy of these approaches, as neither Beyondcell nor scDrug allows for drug combination predictions.

For an unbiased evaluation, we identified the top-15 predicted most effective and least effective monotherapies for each patient according to each method. The results of this comparison are presented in the new **Figure 5**, which illustrates the predictive performances of BeyondCell, scDrug, and scTherapy across the AML patient cohort for the prediction of the measured drug sensitivity score (DSS). In summary, scTherapy showed a significantly better accuracy in distinguishing effective and ineffective treatments, compared to the other methods, in 6 out of 12 samples (50%), $p < 0.05$, Wilcoxon test with FDR correction, while scDrug performed better in 1 sample (8%). Although Beyondcell and scDrug successfully identified several effective treatments, they categorized many drugs with high measured drug sensitivity as ineffective.

These findings are briefly discussed in the manuscript (pages 13-14). We chose not to focus too much on these benchmarking results, since the main focus of our work was on making selective clone-specific predictions and their experimental validation in AML and HGSC patient samples. We note that we compared here the overall drug effect (DSS level, PMID: 37996540), not percentage inhibition of leukemic cells at a given dose, like in our experimental validations, since neither Beyondcell nor scDrug can predict effective doses at which one is expected to see a strong inhibition effect. In one of the AML sample, where scDrug performed better in the DSS comparison, we actually see a much greater separation of effective and ineffective drugs when comparing %inhibition at predicted effective doses with scTherapy (see **Fig. 2b**, Patient 3)

Fig. 5 | Comparison of monotherapy efficacy predictions in 12 AML patients with three methods. Drug sensitivity score (DSS) distributions of the top-15 drugs predicted as (a) the most effective and (b) the least effective monotherapies by each model for individual patients. The predictions were compared using

pairwise Wilcoxon rank-sum tests, with p-values adjusted for the False Discovery Rate (FDR) with the Benjamini-Hochberg procedure. *a model's prediction is significantly different ($p < 0.05$) compared to that of at least one of the two other methods; **a model's predictions show a significant difference ($p < 0.05$) when compared to both of the alternative methods.

The authors should provide some kind of containerization for scTherapy. In its current state, the tool depends not only on common R libraries but also depends on packages which must be downloaded directly from GitHub. Moreover, it is not clear to me which specific versions of each library should be installed. In these cases, conflicts arising from incompatible package versions are common and seriously hamper reproducibility of the results showcased in the paper and the usability of the proposed methodology. Including a docker image or even a conda recipe would help potential users to install and use scTherapy. I was unable to install the required dependencies following the exact instructions laid out in the GitHub repository, most probably because of yaGST requiring an older and incompatible R version compared to my current setup.

Our response: We apologize for the inconvenience when trying to install the R libraries. We have now specified the required versions of each package to mitigate any conflict and improve reproducibility in the GitHub repository (<https://github.com/kris-nader/scTherapy>). We agree that potential conflicts arising from incompatible package versions and challenges during the installation process will decrease the usability of our tool. We have therefore also provided a more user-friendly access to our scTherapy, as suggested, by creating a docker image that ensures compatibility across different environments. The docker image encapsulates all relevant dependencies for running scTherapy, including Seurat, inferCNV, copyKAT, SCEVAN, yaGST, and biomaRt. We specifically highlight these packages as they are most likely to raise installation issues. This will streamline the installation process and enhance the overall usability of scTherapy. The scTherapy users can access the docker containerization on Docker Hub (<https://hub.docker.com/r/kmnader/sctherapy>), and the instructions can also be found on GitHub (<https://github.com/kris-nader/scTherapy>). We have provided these links under the “Data and code” availability section in the revised manuscript (page 21). We have further adapted scTherapy to be compatible with the latest Seurat version 5, with a separate docker provided (https://hub.docker.com/r/kmnader/sctherapy_v5). We have also asked our colleagues to beta-test the R codes and the docker image, and they did not encounter any problems in repeating the results and were able to run the analyses based on the instructions available on the GitHub page. Finally, we have updated the Quick Start guide on GitHub to make it more straightforward and user-friendly, facilitating an even smoother start of scTherapy for all users.

kmnader/sctherapy ☆

By kmnader · Updated 3 days ago

<https://github.com/kris-nader/scTherapy> – infercnv, seurat4, biomaRt, SCEVAN, copyKAT

Image

Overview Tags

No overview available
This repository doesn't have an overview

Docker Pull Command

```
docker pull kmnader/sctherapy
```

Regarding the experimental validation in AML patients, Supl. Table 1. suggests that relapsed, refractory and newly diagnosed patients were considered for the analysis. I wonder about the performance of scTherapy when comparing the newly diagnosed and otherwise treatment naive patient with the relapsed and the refractory patient. Have the authors found any differences regarding the amount of available candidates and the dosages required to reach malignant inhibition in newly diagnosed against refractory patients? Does scTherapy identify a reduction in malignant inhibition for anagrelide in patient 12? In summary, is scTherapy capable of capturing the therapeutic heterogeneity arising from previous treatments? Is the standard of care offered by scTherapy? Does it usually come out? To what extent the results found would improve the standard of care?

Our response: We thank the reviewer for these insightful questions regarding the performance of scTherapy in the AML patient samples. Concerning the comparison between predictions from diagnosis and refractory samples, we conducted such an analysis using AML patient samples 2 and 3, as they are paired samples from the same patient at different disease stages (diagnosis and refractory). This is now clarified in the revised Suppl. Table 1 footnote. In the context of the comparison of samples 2 and 3, it is worth noting that the patient after diagnosis (sample 2) was treated with azacitidine, but this treatment did not induce a significant response, and a relapse (sample 3) occurred soon after the start of another azacitidine-venetoclax treatment regimen. It is therefore challenging to predict expected differences in the drug responses between the two time points (sample 2 to sample 3). Furthermore, since the scTherapy predictions are highly patient-specific, including the other AML patients (either diagnosis, refractory or relapsed) would seriously confound this analysis, and therefore we based these results on this single patient case (no other paired samples with scRNA-seq data were available from the current cohort).

Our analysis revealed no significant difference in the number of drug predictions, nor differences in the predicted overall effective doses either among all the scTherapy-predicted monotherapies (**new Suppl. Fig. 2a**), or in the doses of the common monotherapies predicted by scTherapy for both of the two samples (**Suppl. Fig. 2b**). Interestingly, we found that scTherapy predicted 12 unique drugs to be effective in the refractory sample, but not in the diagnosis sample, consisting of 6 predicted as high-to-moderate responses and 6 with a predicted moderate response (**Suppl. Fig. 2c**). However, since this is only a single patient case with both diagnosis and relapsed sample

available, we do not want to make any strong conclusion based on this case. We note that, in general, scTherapy consistently predicts 20-35 effective drugs per patient sample with a high confidence (and approximately the same number of drugs are identified as effective with a lower confidence), guiding the selection of drugs for validation testing based on these efficacy categories and confidence levels.

Supplementary Fig. 2 | Predicted effective drugs and doses for the diagnosis and refractory samples of the same AML patient. No significant differences were found in the predicted effective doses between the paired AML patient samples 2 and 3 from the same individual. **(a)** Predicted doses of all predicted drugs, difference assessed with two-sample Wilcoxon test. **(b)** Predicted doses of common drugs predicted for both the diagnostic and refractory samples, difference assessed with paired signed rank test. **(c)** Predicted effective drugs in diagnosis and refractory samples of the same individual. scTherapy predicted a total of 27 monotherapies both in the diagnosis and refractory samples, out of which 15 are common between the two samples. Drugs in boldface are those predicted to elicit a high-to-moderate response, while those in italics are expected to produce a moderate response. Epirubicin was predicted in diagnosis and refractory samples with moderate and high-to-moderate responses, respectively.

Regarding patient 12, with anagrelide listed in his treatment history, we confirmed that anagrelide was not predicted by scTherapy, due to its absence in the training set (i.e., this drug is not included in the high-quality overlapping compounds between LINCS 2020 and PharacoDB resources, as defined in Materials and Methods). Furthermore, aspirin and anagrelide have been now omitted from the AML patient treatment table, as these were prescribed for patient 12 to treat a previous malignancy (thrombocytosis). The new **Suppl. Table 2** reflects only those treatments administered for the current malignancy under investigation (AML). Furthermore, we have now focused on scTherapy predictions to treatments given after the sampling, where the use of

scTherapy to assess the predictive outcomes of subsequent treatments resulted in some insightful findings. This new treatment information has been added to the new **Supplementary Table 2** and shown below.

Supplementary Table 2 | Clinical and predicted treatment responses for the AML patients.

Patient	Disease stage	Treatment before sampling (response)	Treatment after sampling (response)	Predicted response to treatment after sampling
1	Diagnosis	-	Hydroxyurea (no)	-
2	Diagnosis	-	Azacitidine (no)	-
3	Refractory	Azacitidine (no) Venetoclax (no)	-	-
4	Diagnosis	-	Hydroxyurea (yes), Cytarabine/Idarubicin (yes), Cytarabine/Daunorubicin (yes), busulfan/cyclophosphamide (yes)	Daunorubicin (HM)
5	Relapse	Hydroxyurea (yes), Cytarabine (yes), Cytarabine/Idarubicin (yes), Research drug treatment (no)	Investigational immunotherapy	-
6	Relapse	Cytarabine/Idarubicin (yes), Mitoxantrone/Cytarabine (yes), Amsacrine/Cytarabine/Etoposide (yes), Idarubicin/Cytarabine/Etoposide (yes), Clofarabine/Cytarabine/Etoposide (yes), Azacitidine (no), Azacitidine/Lenalidomide (no)	Mitoxantrone/Etoposide/Cytarabine (yes), Clofarabine/Cytarabine (no)	Mitoxantrone (HM), Etoposide (M)
7	Diagnosis	-	Azacitidine (no)	-
8	Refractory	Cytarabine/Idarubicin (no)	Cytarabine/Mitoxantrone (no), Azacitidine (no)	Mitoxantrone (HM), cyt:clofarabine (M-LC)
9	Diagnosis	-	Cytarabine/Idarubicin/GF (yes), Cytarabine/Mitoxantrone (yes), Cytarabine (yes)	-
10	Diagnosis	-	Cytarabine/Idarubicin (yes) Cytarabine/Idarubicin/Lenalidomide, Cytarabine/Daunorubicin (yes), Etoposide/Mitoxantrone (yes), Lenalidomide (yes)	Etoposide (M), Lenalidomide (M-LC)
11	Diagnosis	-	Participated in a clinical study	-
12	Refractory	Azacitidine (no), Hydroxyurea (no)	Investigational immunotherapy	-

The columns include clinical treatments before and after taking the sample, as well as the clinical responses (yes/no), based on the percentage of blasts in the bone marrow. Predicted response with scTherapy to the treatments after sampling or drugs with similar mechanisms of action. Drug names in red represent those that were predicted by scTherapy as effective but filtered out due to low confidence of the predictions. Predicted response: HM, High-to-Moderate; M, Moderate; LC, Low confidence. - no scTherapy prediction possible for the patient treatment.

For instance, in patient 10, who was treated with mitoxantrone/etoposide and responded well based on the percentage of blasts in the bone marrow, scTherapy predicted etoposide having a moderate response based on the sample taken before the treatments. Similarly for Patient 6 treated with Mitoxantrone/Etoposide/Cytarabine, scTherapy accurately predicted effectiveness of a subset of the clinical treatment, with high-to-moderate and moderate response to mitoxantrone and etoposide, respectively. In addition, scTherapy was able to predict daunorubicin with high to moderate response in patient 4, which was confirmed to have a good response using the clinical response data from this patient after the treatment.

We note that scTherapy cannot make predictions for some of the commonly administered standard-of-care non-targeted drug treatments, such as cytarabine, azacitidine and hydroxyurea, which broadly affects both malignant and healthy cells. Instead, we focused on prioritizing drugs that effectively inhibit cancer cells and minimally inhibit normal cells - a pivotal aspect of scTherapy. The same applies for the investigational immunotherapies for AML that are not target-specific, and therefore out of the scope of the current version of scTherapy.

Furthermore, there were cases such as patient 11, who underwent consecutive treatments with an investigational compound in an ongoing clinical trial, for which treatment data are not yet available to assess the accuracy of scTherapy treatment prediction. A very late-stage Patient 1 was administered hydroxyurea, considered a palliative care measure, and she exhibited a poor clinical response and passed away a few days later, similar to patient 7, treated with azacitidine.

In addition, there were also instances, where despite the accurate prediction of the administered drug, it was ultimately not included in the final analysis, due to its low confidence score in the conformal prediction (marked in red font in Suppl. Table 2). For scTherapy users who are interested in retaining all the predictions (high and low confidence), the “predict_compounds” function provides an option to include predictions with lower confidence as well.

We note that scTherapy was not designed specifically to capture the therapeutic heterogeneity arising from previous treatments across a cohort of patients, but rather to propose effective and safe multi-targeted treatment options for individual patients at any stages of their disease development (diagnosis, refractory and relapsed stages). We observed a high heterogeneity in the scTherapy-predicted treatments across patients and disease stages. This is expected in such a heterogeneous cohort of AML patients, with various driver mutations and treatment regimens. We have mentioned these aspects in the revised version (page 4).

We made an extensive effort to collect in **Suppl. Table 2** all the treatments from this real-world cohort and manually confirmed clinical treatment responses based on the percentage of blasts in the bone marrow. The current patient cohort is still unfortunately too small and the clinical response data too limited to make any strong conclusions about the match between model predictions and clinical patient responses. However, we expect the patient-specific scTherapy-predicted combinations will lead to more effective and safer treatment alternatives, compared to the current standard-of-care, once these predictions are used in future patient management.

Interestingly, the authors find that there are different clonal subpopulations that share potential treatments. This has been reported in previous articles (see the concept of "therapeutic clusters" introduced by Fustero-Torre et al. (PMID: 34911571)). The Discussion section could be enriched by discussing this point along with the contributions of scTherapy.

Our response: Thank you for pointing this out. We have now described this "therapeutic clusters" concept, used in the BeyondCell method, along with contributions of scTherapy in the revised discussion (page 13).

Consider shortening and structuring the "Discussion" section, as it is quite lengthy (almost resembling the Results section) and sometimes lacks substantial content.

Our response: We have now better structured the Discussion section by adding subheadings and slightly shortening when compared with the original version. We tried to summarize the key messages from the study only in the beginning of the Discussion. However, a number of new discussion points were also added based on the comments raised by the four reviewers.

Minor

In the methods section, the authors state that 10303 overlapping compound-cell lines were considered for training scTherapy. However, it is not clear to me whether all of these drugs are then considered as potential candidates in the tool. That is, which is the universe of compounds that scTherapy is able to propose? A brief statement or a small addition to Fig 1. would be highly informative in this regard.

Our response: We have now clarified in the Fig. 1 caption that scTherapy can indeed propose as potential drugs any of the 3,695 unique compounds that constitute the 10303 compound-cell lines pairs for scTherapy training (page 5). However, the experimental validation is naturally limited to the compounds available in a drug library, and since many of the targeted compounds are ineffective in the tested range of 1-1000 nM, as expected, predictions for less than 1000 compounds are typically available for any given patient sample with various genetic and molecular backgrounds.

In the discussion, the authors claim to have evaluated already existing machine learning approaches, but do not elaborate on specific tools nor publications, citing instead a review. Consider adding a brief statement about competing methods to improve the discussion and to highlight how scTherapy fits.

Our response: We have now added brief statements about the competing methods, BeyondCell and scDrug, and how they compare to scTherapy, as well as limitations of other methods, as summarized in a recent review paper of single-cell level drug response prediction methods [PMID: 38109840] (pages 13-14).

The authors claim that their approach is a widely-applicable strategy to prioritize personalized therapeutic candidates. As far as I understood, scTherapy relies mostly on the gene expression profile of each malignant population to find compounds able to reverse the disease state. However, I wonder about tumor types for which the most predictive feature for a given targeted therapy is actually a point mutation. For instance, melanomas harboring the BRAFV600E mutation benefit from BRAF inhibitors such as vemurafenib. However, studies of the predictive power of gene expression in this context have showcased that gene expression has little relative importance for predicting drug response, being the presence of the mutation the most useful feature. See: <https://depmap.org/portal/compound/VEMURAFENIB?tab=predictability>.

Our response: We agree that for tumor types or patient cases for which the most predictive feature for a given targeted therapy is actually a point mutation are currently beyond the scTherapy approach. This is because we wanted to make the first version as practical and easy-to-apply as possible by requiring only the scRNA-seq count matrix of the cancer sample. We have now listed this as one of the directions for future developments, using the BRAF-V600E point mutation as example, toward future ScTherapy versions that make use of multi-omics data (page 14). Panel mutation data are indeed many times available from patient samples, and often carry additional information compared to the CNV data, and hence would be the next data type to be incorporated into the scTherapy approach.

However, we also want to emphasize that for many tumor types and targeted drugs, there are no point mutation markers available; even for vemurafenib, the BRAF-V600E mutation is often not predictive marker in other tumor types, including colorectal cancer, where activation of EGFR signaling mediates a bypass signal for BRAF-mutated colorectal cancer cells to become non-responsive to BRAF inhibitors such as vemurafenib (PMID: 22281684 and PMID: 22448344). There is actually a rather small number of specific point mutations that have a strong predictive power for predicting a few very specific drugs (such as BRAF mutations vs. (monomeric) BRAF inhibitors). However, for broader sets of drugs with different mechanisms of action, the transcriptomic data is often a more powerful predictor. This was found, for instance, in a DREAM Challenge study that used various omics data, including gene expression and point mutational patterns, to predict drug responses and showed that the gene expression data overall predicts drug responses the best:

Costello JC, Heiser LM, Georgii E, Gönen M, Menden MP, Wang NJ, Bansal M, Ammad-ud-din M, Hintsanen P, Khan SA, Mpindi JP, Kallioniemi O, Honkela A, Aittokallio T, Wennerberg K; NCI DREAM Community; Collins JJ, Gallahan D, Singer D, Saez-Rodriguez J, Kaski S, Gray JW, Stolovitzky G. A community effort to assess and improve drug sensitivity prediction algorithms. *Nat Biotechnol.* 2014 Dec;32(12):1202-12. doi: 10.1038/nbt.2877.

Reviewer #2, expertise in ovarian cancer organoids (Remarks to the Author):

This is an interesting and ambitious study for the development of new tools for improving data-driven decision-making approaches in oncology. The authors developed a new computational model (Light GBM) and trained it on available data sets containing data on genome-wide transcriptomic profiles from 167 cell lines (LINCS2020) project and added data on viability responses from the Pharmaco DB platform.

Our response: We thank the reviewer for appreciating our work, and the ambitious approach for improving data-driven decision-making approaches in oncology. We have addressed all the comments and provide our point-by-point responses below. All the text changes in the revised manuscript are highlighted in yellow.

The experimental part of the study is mainly focused on testing predicted treatments in the AML patients group. The new sequencing data set from 12 bone marrow samples has been produced and comprehensive in vitro validation performed. In line with this and supporting the basic premise of the study to select optimal multitarget therapy to address cancer heterogeneity and defined personalized treatment in each case authors identify combined top effective and selective compounds for each patient. In vitro data includes a comparison of single effective single agents vs. noneffective agents, and also tests of combinatorial two-drug treatments for 4 patients. The figures Figures 2b.d.e. convincingly demonstrate the ability of the model to pick effective drug combinations for AML treatment

In the case of ovarian cancer experimental data is limited and significantly less convincing. While the wish of authors to extend the application field of their tool to solid malignancies is understandable, the presented data has significant weaknesses and limitations, to support the overall conclusions of the study.

Our response: We agree that the originally reported ovarian cancer experimental data was rather limited compared to the AML use case; the reason being that we wanted to showcase one solid tumor application case and to extend the applicability field of our method. We have now significantly extended the experimental validation of the ovarian cancer use case, as detailed below. We also refer to our response to the first comment of Reviewer 1 that showcases additional applications in other solid tumors using publicly available single-cell data.

Some of the major issues are:

- **The validation is performed only with one organoid line from a single patient. This conflicts with the main claim of the study to select a treatment for successful targeting of cancer heterogeneity. To achieve that one would have to provide evidence by testing predicted combinations on several clonal lines from a single patient included in the experimental setup. Independently of heterogeneity issue, single patient validation strategy is inadequate.**

Our response: We agree that including more samples and patients in the analysis will make the application to predicting clonal-level drug response in genetically heterogeneous solid tumors more convincing. We therefore analyzed the available scRNA-seq transcriptomics data from our HGSC cohort and generated predictions for selective drug sensitivity in two additional HGSC patients, where we established organoids and stromal cell cultures to test the tumor-selectivity of the predicted treatments on the same patient cells. **Figure 3** is now updated accordingly, see below, and we have summarized these new results on pages 7-8 of the revised manuscript. The experimental details are described in the Materials and Methods section (pages 20-22).

Fig. 3 | Experimental validation in ovarian cancer patient-derived tumor organoids. (a-c) UMAP projection of the scRNA-seq transcriptomic profiles for the three HGSC patient samples, using standard Seurat integration workflow, where cell types were identified with ScType (left panel). Expression of the PAX8 and EPCAM markers, effectively separating tumor cells from the other cell populations (middle panels). Cell inhibition differences between the patient-derived organoid cancer cells (PAX8+) and non-cancerous normal cells (PAX8-) for the 18 predicted multi-targeting drugs (right panel). The predicted effective doses are indicated in parentheses (μM), and the dotted vertical lines indicate 50% inhibition. The error bars represent SEM, based on three replicates of organoid treatments and curve-fitting in PAX8- cells. For patients 2 and 3 both organoids and the stromal cell cultures were available at the cell numbers sufficient for single-drug sensitivity and selectivity testing, whereas for Patient 1 the PAX8+ tumor cells originated from the patient organoid and PAX8- normal cells were available from additional Patients 4 and 5 (**Suppl. Table 3**). (d) Representative immunofluorescent image of treatment-naive tumor organoids from the Patient 1 sample. (e) Statistical comparison of the treatment responses between PAX8+ and PAX8- cells across three HGSC patient samples with the Wilcoxon test.

We have now clarified our approach to predicting multi-targeting monotherapies exclusively for the HGSC patient samples, rather than making combination predictions, like in the AML patient samples. This decision was informed by the scRNA-seq clustering analysis presented in **Fig. 3a-c** (left panel), which reveals a minimal proportion of cancer cell populations. The limited size of the cancer cell populations constrains the identification of cancer subclones for combinatorial therapy prediction across these HGSC samples. Consequently, we have opted to predict monotherapies by considering all cancer cells within a patient sample as a unified malignant entity, without differentiation between subclones. This is now clarified both in the Results (pages 7) and Materials and Methods sections (pages 18 and 20).

We further acknowledge this limitation in the Discussion section (page 15), highlighting the impact of cell population size on the resolution of genomic variation detection that is critical for subclone differentiation: “In samples where scRNA-seq detects only a small fraction of cancer cells, further dividing this subset becomes unreliable, as it would increase noise rather than distinguishing cancer-related signals. Given this challenge in our HGSC samples, we chose to predict responses to single-agents, instead of drug combinations, relying on comprehensive transcriptomic profile of the cancer cells (instead of smaller sub-clones). Therefore, we advise users of scTherapy to consider the limitations of cell population size and the potential for increased noise when attempting to delineate subclonal populations. In such cases, the analysis of the overall transcriptomic landscape of malignant cells may offer a more feasible and effective strategy for identifying multi-targeted monotherapies, rather than subclone-specific drug combinations.”

Since this is a “n=1” precision oncology approach (i.e., the predictions and experimental testing is carried out for each patient separately), the cohort size does not determine the significance of the results. We hope that these additional organoid HGSC cases will make the validation strategy more adequate. We also refer to our response to the first comment of the Reviewer #1 that show additional applications in other solid tumors using publicly available single-cell data, and their comparison to the predictions of the HGSC cases. We believe these additional results have significantly enhanced the value of the approach across a wide range of tumor types.

• In the case of ovarian cancer the authors decided to perform only a comparison of a single treatment between cancer and noncancer cells (fibroblast) citing limitations in the cell number of organoids. This explanation is not conceivable. PDO organoids, if properly cultivated, are robust models and should be easily expandable to a multiwell format and multiple plates which would easily allow multi-drug combinations testing. The source publication authors cite and where they obtained organoids from does show stable long-term growth so it is not understandable why there should be any limitations.

Our response: We apologize for the poor wording in the original version. It is indeed not the number of organoid cells but rather the replicative arrest of the non-transformed cells that poses a bottleneck in terms of the cell number in this setting. The success of expanding the matched pairs of both organoids and the stromal compartment from the tumor sample of the same patient varies. Fortunately, during the revision phase, we managed to secure two additional patients for

which both organoids and the stromal cell cultures were available at the cell numbers sufficient for single-drug sensitivity and selectivity testing (please see the new **Fig. 3** and **Suppl. Table 3**).

• **Also, it is not clear why a comparison of effective vs. ineffective treatment, similar to Fig 2b is omitted in the case of the ovarian cancer model. This is should be straightforward and doable.**

Our response: We concur that a comparison between effective and ineffective treatments is indeed straightforward, and we initially sought to include such an analysis for the ovarian cancer models, akin to what was presented in **Figure 2b** for the AML patients. The long-term expansion of the PAX8-negative stroma-derived cells appeared a bottleneck for the feasibility, as unlike the organoids, these non-transformed cells undergo cell division arrest within just 2-3 passages, limiting the number of assays that could be performed. Therefore, given the finite quantity of stromal cells derived from the ovarian patient samples, we prioritized spending the stromal cell amount for testing the model predictions (**Figure 3a-c**, right panel). We found that 31 of the 54 predicted effective treatments (57.4%) resulted in greater than 50% inhibition of the PAX8+ tumor cells, which is coherent with the findings in AML patient samples (**Figure 2b**), where the expansion of non-cancer cell population did not limit the experimental validation.

• **The usage of fibroblasts as “normal cells” is not ideal as it essentially involves a comparison of 2D vs. 3D cell culture models which is likely a confounding factor. It would be much better if the authors would make a comparison to healthy organoids from the same patient when ovarian cancer is taken. I understand that this point would be difficult to address without obtaining fresh samples, nevertheless, the issue needs to be addressed or at least in-depth discussed beyond brief mention in the current version of the manuscript.**

Our response: We agree with the reviewer that including control cell models that closely match the growth conditions of the cancer cells would improve the validation strategy for scTherapy approach for solid tumors. Unfortunately, we could not obtain fresh samples for this revision, but we will definitely try to include those in the future applications. We note this was the first study, where we developed the scTherapy approach, initially tested using the AML patient samples, and the ovarian cancer case study came later when we wanted to test its performance also in the solid tumors. We have now expanded the discussion of this issue on pages 15-16:

“The *ex vivo* validation strategy for the selective predictions would also benefit from incorporation of different types of control models, ideally closely matching the growth conditions of the cancer cells, which would minimize confounding factors, such as medium composition, matrix requirements, or growth dynamics of the model. In this context, adding different types of “healthy controls”, for instance, orthotopic non-transformed cells together with PBMC or iPSC organoids from the same patient, would offer alternative validation strategy in solid tumors.”

• The description of the processing of scRNA data of ovarian cancer correctly postulates that PAX8 expression can be used as a discriminatory marker to distinguish cancer vs. noncancer cells in the tissue as PAX8 is uniformly expressed in cancer tissue. Organoids represent only the epithelial compartment and thus should be strongly PAX8 positive. Authors state that organoids contain only cancer cells. Still, Figure 3c of treatment naïve tumor organoids shows an image that contains multiple cells that are PAX8 negative. What is the origin of these cells? Are all cells Epcam positive?

Our response: We thank the reviewer for this important note. We would like to note that the statement that the organoid cultures contain only cancer cells is based on the results of the whole-genome sequencing of the organoid cultures, profiling of their copy-number variation, and determination of the tumor cell purity based on the sequencing-based factors (Senkowski et al, 2023; ref. 27), where it was found that organoids were characterized by high tumor purity (99.2% \pm 1.1%). Moreover, TP53 mutation analyses, revealing a Variant Allele Frequency (VAF) of 1.0 in all organoids, confirms that organoids comprise only cancer cells (Senkowski et al, 2023; ref. 27). This is now clarified in the revised version (page 20).

The mRNA expression of PAX8, as well as Epcam and MUC16, was used to discriminate the cancer cell populations in the scRNAseq analysis. We have now also shown expression of EPCAM and PAX8 marker genes in **Figure 3a-c** for each HGSC patient. We have added a western blotting analysis of the EpCam and PAX8 protein expression (new **Supplementary Fig. 6a**, see below), which confirms that both EpCam and PAX8 proteins can be detected in the tested organoid cultures, but not in the stromal compartment samples. We also performed the PAX8 protein expression quantification based on the immunofluorescence imaging of the organoid culture included in the original version (Patient 1), and in one control HGSC cell line, KURAMOCHI (new **Supplementary Fig. 6b-d**). This quantification indicated that 100% of KURAMOCHI cells and 84% of the organoid nuclei had a signal level that exceeded the background staining (thresholded as mean \pm 3SD for the control staining intensity with only 2ry conjugates), suggesting PAX8-positive status of the model (new **Supplementary Fig. 6b-d**).

However, both in the organoids and the cell line, we observed variability in the detected protein levels (**Supplementary Fig. 6d**). This observation aligns with the mRNA expression analyses presented in this work as well as in the previous studies of the same patient cohort (refs. 25 and 27), where immunofluorescence analysis suggests that although PAX8 serves as a highly selective marker for HGSC cancer cells, not all HGSC cells express high levels of PAX8 protein. This variation underscores the phenotypic heterogeneity retained in the organoid cultures. The origin of the cells with a low level of PAX8 expression is an interesting research question in itself. While exploring this in more detail is outside the scope of our current study, one potential explanation could involve cell cycle-dependent expression of PAX8, as suggested by previous studies that link PAX8 to cell cycle regulation:

1. Qiu, P., Jie, Y., Ma, C. et al. Paired box 8 facilitates the c-MYC related cell cycle progress in TP53-mutation uterine corpus endometrial carcinoma through interaction with DDX5. *Cell Death Discov.* 8, 276 (2022). <https://doi.org/10.1038/s41420-022-01072-8>

2. Qiu, P., Jie, Y., Ma, C. et al. Paired box 8 facilitates the c-MYC related cell cycle progress in TP53-mutation uterine corpus endometrial carcinoma through interaction with DDX5. *Cell Death Discov.* 8, 276 (2022). <https://doi.org/10.1038/s41420-022-01072-8>

Supplementary Fig. 6 | Experimental analyses in the HGSC patient samples and organoids. (a) Immunoblotting for PAX8 and Epcam proteins in the stroma-derived and organoid samples from the representative individual HGSC patients. **(b)** Quantification of the imaging data for PAX8 nuclear expression in Patient 1-derived HGSC organoids (related to **Fig. 3e**), and in one control HGSC cell line, Kuramochi. Anti-rb-555, secondary goat-anti-rabbit-Alexa555 antibody conjugates. **(c)** Representative images of the immunofluorescence imaging of the nuclear PAX8 expression for the Patient 1 organoids. **(d)** Statistical

analysis of the imaging data presented in panels b and c. The fraction of PAX8+ objects correspond to the nuclei with PAX8-555 signal higher than mean+3SD for the respective control without PAX8 primary antibodies.

Minor point

- **The abstract is very general and should be amended to be more specific and closely related to the experimental data.**

Our response: We have now made the abstract more specific and related to the experimental data (page 1). We note that the current abstract is already at maximal word count of 200 words.

Overall this is an interesting study potentially of interest for the broader research and medical community interested in drug treatment predictions and the development of personalized therapies. However, the prediction model requires more robust wet lab testing. In particular for its application for the treatment of solid malignancies, in this case, ovarian cancer. Expanding the number of tested organoid lines beyond n=1 and testing of combinatorial treatments as well as comparison of effective vs non effective drugs should be prioritized to adequate experimental validation data to support the overall conclusions of the study.

Our response: We hope these additional results, both the new wet-lab validations in ovarian cancer organoids (please see our responses above) and using publicly available pan-cancer single-cell data (please see our response to the first comment from Reviewer #1) demonstrate the versatility and scope of the method in multiple tumor types and single-cell datasets.

Reviewer #3, expertise in clonal evolution and bioinformatics (Remarks to the Author):

Advanced malignancies often exhibit complex intratumor cellular diversity and require therapies that target multiple aspects of the disease. However, the high number of potential drug/dose combinations makes identifying patient specific treatments challenging. In this work, the authors develop a machine learning approach which exploits single-cell transcriptomic data to prioritize personalized multi-targeting treatments for patients with hematological cancers or solid tumors. The proposed framework was applied to a dataset comprising single-cell RNA-seq data for 12 AML patients and to single-cell data for 3 ovarian cancers and identified patient-specific treatment combinations targeting various biological pathways. Some of the results were experimentally validated.

Overall, I believe personalized treatment to be a crucial aspect of cancer care, making this a significant topic. However, I have a few concerns I would like the authors to address.

Our response: We thank the reviewer for appreciating the importance of our work for personalized cancer treatment. We have addressed all the comments and provide our point-by-point responses below. All the text changes in the revised manuscript are highlighted in yellow.

1) The manuscript is generally well-written. However, I recommend some improvements to the structure, particularly within the Introduction, Results, and Discussion sections. In the current draft, the initial portion of the Introduction appears to be primarily focused on hematological malignancies, which is fine. However, later in the paper, the authors discuss the significance of their approach in the context of practical limitations for a broad drug testing on solid tumors. In support of this, the authors present a case study on ovarian cancer.

To enhance the manuscript's clarity and relevance, I suggest expanding its framing to encompass a broader context. Specifically, it would be beneficial to provide a more comprehensive introduction that highlights the broader implications of the study, not only for hematological malignancies but also for solid tumors. Emphasizing the significance of their approach across various cancer types, with a clear rationale, will help readers better appreciate the relevance and potential impact of the research.

Our response: We have now extended the introduction to provide a broader context, including state of the art of single cell profiling in cancer research, and the importance of TME, not only for hematological malignancies but also for solid tumors, emphasizing the significance of our approach across various cancer types (page 2). The introduction also describes earlier studies in melanoma, ovarian and colorectal cancers that have demonstrated that specific characteristics of the tumour immune microenvironment (TME) can, to some extent, predict patient's clinical outcome, before going to the specific case study on patients with high-grade serous ovarian carcinoma (HGSC). We have also performed additional analyses to showcase applications in other solid tumors using publicly available single-cell data (pages 8-9 and new **Fig. 4**). We believe these additional results in pan-cancer single-cell data demonstrate the versatility and scope of the method in multiple tumour types and single-cell datasets, underscoring its potential in advancing personalized cancer therapy.

2) In my opinion, the Discussion section is quite long and contains introductory concepts, such as the limitations of the current state-of-the-art machine learning approaches for predicting anticancer drug combinations. I believe it would be more effective to present these introductory notions in the Introduction section instead.

I would recommend that the authors perform a reorganization of the manuscript to provide a more comprehensive introduction to the problem. This should encompass both the biological/wet lab aspects of the topic and the bioinformatic approaches/literature they intend to improve and why.

Our response: We agree and hope the reorganization and additions in the Introduction make it now easier to follow the remaining text, both the biological and wet lab aspects of the topic, and

the bioinformatic approaches we intend to improve and why. We have also now better structured the Discussion section by adding subheadings, as well as we have slightly shortened it compared with the original version. We tried to summarize the key messages from the study only in the beginning of the Discussion. However, we chose to keep the lengthier description of the limitations of the current state-of-the-art machine learning approaches for predicting anticancer drug combinations in Discussion, just to keep Introduction shorter and more targeted to the key aims of the work. Furthermore, a number of new discussion points were also added based on the comments raised by the four reviewers, including comparison against current state-of-the-art machine learning approaches for predicting anticancer drug treatments, and their limitations.

Moreover, I would ask the authors to better discuss the novelty of ScTherapy with respect to previously published approaches.

Our response: We have now added brief statements and novelty of scTherapy with respect to previously published approaches, BeyondCell and scDrug, and how they compare to scTherapy, as well as limitations of other methods, briefly in Introduction (pages 2-3) and lengthier in Discussion (page 13). In particular, we refer to a recent review paper of single-cell level drug response prediction methods [PMID: 38109840], which summarized the key current limitations of the existing methods: (i) they cannot predict drug combination responses, rather focus on single drug effects; (ii) they do not attempt to predict drug dosage, which is critical for translational applications; and (iii) they have not been experimentally evaluated to truly evaluate their practical utility. Our work addresses all these limitations of the current methods for single-cell level drug treatment response prediction cancer applications (page 13).

3) The ScTherapy framework includes a series of preprocessing steps, starting with raw scRNA-seq counts derived from samples obtained from cancer patients. These preprocessing steps are designed with two primary objectives: to establish distinct clusters of cells, distinguishing between normal cells and clones/subclones within the dataset. And, to identify and compile a list of differentially expressed genes that characterize the specific clones with respect to the normal cells.

I have two concerns regarding this part, which I believe the authors should carefully address in their manuscript. First, it would be valuable to understand the impact of the stability of the detection process differentiating between normal cells and the various clones. Given that ScTherapy relies on differentially expressed genes to characterize each clone relative to normal cells for drug prediction, it's crucial to assess how uncertainties in clone detection might impact the method's reliability. A discussion on the robustness of this process and potential strategies to mitigate uncertainties would enhance the manuscript's rigor.

Our response: We appreciate the comment regarding the stability of the clone detection process, especially in the differentiation between normal cells and various genetic clones, and its critical role in assessing the reliability of scTherapy treatment predictions. To address this important question, we used our AML patient cohort and randomly removed cells in these mixed samples

to investigate how consistent the detection of malignant and healthy cells remain under varying cell sub-samples, thus contributing to a systematic understanding of the stability of the detection process. The percentage of correctly annotated cells was used to measure how well the cell type annotations after the cell removal compared to the original annotation.

As illustrated in the new **Supplementary Fig. 10** (see below), scTherapy detection procedure maintains a surprisingly stable performance in most samples, even when as large proportion as 75% of cells are removed. This suggests its robust performance on diverse datasets with different proportions of healthy and malignant cells, and relatively stable capability to capture relevant malignant and healthy signatures of mixed cell samples. As expected, when more than 75% of cells are removed, the performance becomes unstable. However, comparison of the scTherapy detections to the expected values from beta-binomial distribution under the null hypothesis of random cell assignments reveals scTherapy's significant relative performance (**Suppl. Fig. 10**). We have added these technical stability results into the revised manuscript (page 17).

Supplementary Fig. 10 | Stability of the subclone detection process in distinguishing normal and malignant cells. In this sub-sampling analysis, 25%, 50%, and 75% of cells were first randomly removed at increments of 25% from the 12 AML patient samples (separate traces). Thereafter, using increments of 5% from 75% to 95%, we studied the breaking point of the clone detection procedure. After each removal of cells without replacement, Step 3 of ScTherapy was rerun. The y-axis illustrates the percentage of correctly assigned cells before and after the cell removal. The expected curve illustrates the anticipated trend when modeling the percentage of correctly assigned cells under a beta binomial distribution. The shaded areas in blue depict the 95% quantile for each point estimate, providing insight into the variability of the expected values under random cell assignment. The beta-binomial distribution was selected to account for variations in the healthy/malignant cell classification success rates, capturing the varying accuracy as the proportion of these cell types changes due to the removal of different cell percentages.

We hope these additional analyses improve the manuscript by providing insights into the impact of the detection process stability and reliability. We also note that the performance and robustness of the published approaches used in the detection pipeline, including copyKat and SCEVAN, have been investigated in the original studies (PMID:33462507, PMID:36841879).

Second, considering the objective of personalized medicine and the substantial heterogeneity in cancer, the authors should provide insights into how ScTherapy could be adopted to offer treatment suggestions for new patients in relation to the detected clones for which candidate therapies are designed. Exploring methods to generalize ScTherapy predictions beyond the studied cases and addressing the challenges of extending personalized treatment recommendations to diverse patient populations would strengthen the manuscript's applicability and relevance in clinical contexts. Should the absence of genetic information be considered a limitation? I recommend that the authors discuss this topic in their manuscript.

Our response: To address this important comment, we have now added new application cases in three additional solid tumors, using publicly available single-cell data, and compared these treatment predictions to those identified for the AML hematological case study and the HGSC patient cohort. Each of these cohorts include diverse patient populations in terms of genetic background, disease stage, treatment history and tumor type. Please see our response to the first comment of Reviewer #1 for the detailed results. This comparison shows that scTherapy can predict both recurrent and patient-specific clinically relevant multi-targeting treatments for various tumor types. We have added these new results into the revised manuscript (pages 8-9 and new **Fig. 4**) We think these additional results in the pan-cancer single-cell data demonstrate the versatility and scope of the method in addressing the diverse and complex landscape of cancer treatment predictions, paving the way for identification of targeted and effective therapeutic strategies across a range of tumor types.

As for the absence of genetic information, we want to point out that the detection of clones is based on both transcriptomic and inferred CNV profiles, both extracted from the scRNA-seq data. We agree that for tumor types or patient cases for which the most predictive feature for a given targeted therapy is actually a point mutation are currently beyond the scTherapy approach. This is because we wanted to make the first version as easy-to-apply as possible, by requiring only the scRNA-seq count matrix of the cancer sample. We have now listed this as one of the directions for future developments, using the BRAFV600E point mutation as example, and for next ScTherapy versions that make use of multi-omics data (page 14). Panel mutation data are indeed many times available from patient samples, and often carry additional information compared with CNV data, hence would be the next data type to be incorporated into the scTherapy approach

4) I appreciate the authors' effort in constructing a drug-response database and in employing advanced machine learning techniques to address the drug prediction task. Nevertheless, in the current manuscript, there is no quantitative assessment regarding the performance of their method. While I understand that this might not be straightforward, I

believe it's essential to include quantitative data, possibly through simulations, to validate the effectiveness of both the method and its current implementation.

Our response: The originally reported AML and HGSC application cases included quantitative statistical assessment of the scTherapy responses when compared to the experimental validations (please see **Figs. 2 and 3**). We believe that experimental validations are better than simulations, and it is not clear how to perform realistic simulations in the personalized treatment setting. Additionally, we have now quantitatively compared the scTherapy predictions against existing computational methods in the AML patient cases (please see our response to the third comment of Reviewer #1 and new **Fig. 5**). We believe these experimental and statistical evaluations provide quantitative assessment of the confidence of our prediction approach, like discussed on page 11: “The quantitative performance evaluation (repeated cross-validation and experimental validations), together with the confidence scoring (conformal prediction), enables medical professionals to decide when and how to use the model to guide clinical decision making.”

5) As a final comment, I would like to discuss the authors' claim regarding the particular significance of their approach in the context of drug prediction for solid tumors. I do not argue against this statement. However, I would like to notice that the current experimental results presented in the manuscript are predominantly emphasized on hematological cancers. While I understand that conducting experiments on solid tumors, such as ovarian cancer, might be resource intensive, however I believe that a more extensive set of experiments in this context could significantly enhance the manuscript's value and significance.

Our response: We agree that the originally reported ovarian cancer experimental data was rather limited compared to the AML use case; the reason being that we wanted to showcase one solid tumor application case and to extend the applicability field of our method. We have now significantly extended the experimental validation of the ovarian cancer use case, as detailed in response to the comments from Reviewer #2. We also refer to our response to the first comment from Reviewer #1 that show additional applications in other solid tumors using publicly available single-cell data. We believe these additional results have significantly enhanced the value and significance of the approach across a wide range of cancer types.

Reviewer #4, expertise in AML functional genomics and clonal evolution (Remarks to the Author):

In this paper, the authors use scRNA-seq and machine learning to identify combinatorial therapies that could potentially target multiple clones in a single patient's cancer. This (semi)automated approach would have excellent potential for clinical delivery, although significantly more validation would be required. Nevertheless, the approach is innovative and goes in the direction that is needed to ultimately provide curative therapies for cancer patients.

This is a very well written paper addressing an important problem.

Our response: We thank the reviewer for appreciating the importance of our work for personalized cancer treatment. We have addressed all the comments and provide our point-by-point responses below. All the text changes in the revised manuscript are highlighted in yellow.

My main issue is that in Figs 2e and 3d, there is no statistical comparison to show whether the differences between normal and malignant cells are significant or not. Given that there are replicates (although details of this are unclear), the authors should derive p-values and then revise the numbers estimating the proportions of samples that identify combination therapies that could be efficacious as well as the proportions that would be differentially targeted compared to normal cells.

Our response: As suggested, we have now computed p-values for the comparisons that show whether the differences between normal and malignant cells are significant or not (see **Figure 2e** caption and new **Figure 3e**). Accordingly, we have revised the numbers estimating the proportions of samples that identify combination therapies that are selectively sensitive towards the malignant cells (pages 5-7 and 11).

The other issue is that while it makes sense the drugs and dosages provided are in reference to the malignant cells, this is not necessarily what would be reflected in the clinic. The authors should discuss the point that dose adjustments are often required depending on age, comorbidities and other issues.

Our response: Thank you for pointing out the important translational point. We have discussed this in the revised version (page 13).

REVIEWER COMMENTS

Reviewer #1 (Remarks to the Author):

The authors have addressed and solved all the questions that were posed to them. The new results and benchmarking incorporated have considerably enriched the scientific value of the study. In my opinion the manuscript has also improved considerably with the new modifications and is now more direct and clearer in explaining the main contribution of the paper. It has also enriched and deepened its content in the introduction and discussion. In addition, the software has been reimplemented and is now much more versatile, easily installable and executable and has been documented in more depth.

Reviewer #1 (Remarks on code availability):

The code downloads and executes well following the documentation provided by the authors.

Reviewer #2 (Remarks to the Author):

The authors of the study „Single-cell transcriptomes identify patient-tailored therapies for selective co-inhibition of cancer clones” put considerable effort into improving the Manuscript and have rewritten major portions of the text for clarity. While the new bioinformatics approach is without a doubt interesting and promising and would be of interest for drug development research, the experimental validation regarding the ovarian cancer model remains insufficiently convincing.

Out of several major concerns raised during the first revision, the authors addressed in the wet lab only one, regarding a single line used for validation and added experiments with two further organoid lines. Notably these lines are generated from ascites and not from solid tumor deposits, which introduces further variable in the system. Nevertheless, the trend of the higher sensitivity of cancer cells to the selected treatments in comparison to stroma appears to hold up and is relatively consistent.

Importantly, while the headline of the study is focused on the heterogeneity of the cancer and selection of effective treatments for specific clones this aspect is not addressed in ovarian cancer as tissue sequencing data contained too few cancer cells to identify clonal subpopulations. Then in an additional step of methodological simplification, authors still did not include experiments with effective vs ineffective treatments in ovarian cancer models citing issues with expandability of stroma cells.

Having in mind the fact that fibroblast culture is entirely different from than 3D organoid model, which by itself can influence drug response, observed differences though statistically significant. are still not sufficiently convincing on the conceptual level.

Discussion acknowledges this concern, but this does not amend the fact that experimental validation is yet to include testing of predicted patient-specific drug sensitivity responses in ovarian cancer organoid model. Authors could, for example, simply perform prediction for

the individual cancer samples and then test the top effective treatments only in the organoid lines to assess response variability among 3 donors when subjected to specific treatments.

If they can show that their bioinformatics pipeline can identify effective treatments in individual lines, that could considerably improve the data and increase the potential impact of the publication to extend conclusions to solid tumor malignancies.

Minor point: The scalebar is missing in Figure 3e. These are new pictures presumably to avoid discussion regarding PAX8 negative cells, but now magnification is smaller, thus pictures are somewhat less informative regarding organoid morphology.

Reviewer #3 (Remarks to the Author):

The revised version of the manuscript addresses most of my comments, and overall, I believe it has improved a lot in terms of readability and focus. I also appreciated the authors' effort to perform new analyses to extend the scope of the method beyond hematological malignancies.

My remaining concern is related to the limited quantitative evaluation of the performance of the authors' method and its quantitative comparison and improved significance with respect to other state-of-the-art approaches in the field.

As stated in my original review, I understand that this is not straightforward, and I appreciate the new paragraph included by the authors in the discussion section where they conduct a review of the literature and mention possible competing methods.

I do not dispute the performance of scTherapy. However, I believe demonstrating the tool's performance is important to establishing its significance.

I do not think this part should be mentioned only in the discussion, and I would ask the author to improve the visualization of Figure 5, as its current version does not allow for easy comparison among the methods.

Reviewer #3 (Remarks on code availability):

I was only capable of performing a quick assessment, and the R repository seems ok.

Reviewer #4 (Remarks to the Author):

My critiques have been addressed

REVIEWER COMMENTS

Reviewer #1 (Remarks to the Author):

The authors have addressed and solved all the questions that were posed to them. The new results and benchmarking incorporated have considerably enriched the scientific value of the study. In my opinion the manuscript has also improved considerably with the new modifications and is now more direct and clearer in explaining the main contribution of the paper. It has also enriched and deepened its content in the introduction and discussion. In addition, the software has been reimplemented and is now much more versatile, easily installable and executable and has been documented in more depth.

Reviewer #1 (Remarks on code availability):

The code downloads and executes well following the documentation provided by the authors.

Our response: We thank the reviewer for appreciating the efforts we have made to address all the comments, and to improve our manuscript and the software implementation.

Reviewer #2 (Remarks to the Author):

The authors of the study “Single-cell transcriptomes identify patient-tailored therapies for selective co-inhibition of cancer clones” put considerable effort into improving the Manuscript and have rewritten major portions of the text for clarity.

Our response: We thank the reviewer for appreciating the efforts we have made to improve our manuscript and address the comments.

While the new bioinformatics approach is without a doubt interesting and promising and would be of interest for drug development research, the experimental validation regarding the ovarian cancer model remains insufficiently convincing.

Out of several major concerns raised during the first revision, the authors addressed in the wet lab only one, regarding a single line used for validation and added experiments with two further organoid lines. Notably these lines are generated from ascites and not from solid tumor deposits, which introduces further variable in the system. Nevertheless, the trend of the higher sensitivity of cancer cells to the selected treatments in comparison to stroma appears to hold up and is relatively consistent. Importantly, while the headline of the study is focused on the heterogeneity of the cancer and selection of effective treatments for specific clones this aspect is not addressed in ovarian cancer as tissue sequencing data contained too few cancer cells to identify clonal subpopulations.

Our response: We appreciate the reviewer's insightful comments on the experimental validation of our approach. We have acknowledged the potential variability introduced by our use of ascites-derived organoid lines of ovarian cancer patients, a decision primarily driven by sample availability. In comparison to the AML patient-based validations, the HGSC case study turned out to be much more challenging in terms of both sample availability and drug treatment testing in the patient-derived organoid lines to validate the model predictions.

Despite the inherent differences between ascites-derived samples and solid tumor deposits, our scRNA-seq data analysis successfully identified both cancerous and non-cancerous cells within the ascites-derived samples. This allowed us to perform differential expression analysis between cancer and normal cells, and consequent experimental validation of the treatment predictions in organoids derived from the same ascites-sourced cancer cells. Such alignment ensures consistency between our scRNA-seq analysis and experimental validation.

Given the limited number of cancer cells detectable in the scRNA-seq profiles of the patient samples, we indeed chose to focus on predicting responses to single, yet multi-targeted drugs. This is because the tissue sequencing data in ovarian samples provided insufficient cancer cells to reliably identify clonal subpopulations, although we were able to consistently identify subclones in AML patient samples. We have discussed this limitation in our revised manuscript, and now also included this description in the GitHub user guide for transparency (<https://github.com/kris-nader/scTherapy/blob/main/README.md#notelimited>).

As the reviewer pointed out, the experimental validations of our single-drug response predictions in the ovarian patient samples consistently demonstrated a higher sensitivity of cancer cells to the model-identified treatments compared to stromal cells. We are grateful for the reviewer's recognition of the importance of these findings, which we also find rather impressive, given that only the scRNA-seq count data was used as input for the machine learning model to predict such patient-specific and cancer-selective treatments.

Then in an additional step of methodological simplification, authors still did not include experiments with effective vs ineffective treatments in ovarian cancer models citing issues with expandability of stroma cells. Having in mind the fact that fibroblast culture is entirely different from than 3D organoid model, which by itself can influence drug response, observed differences though statistically significant, are still not sufficiently convincing on the conceptual level. Discussion acknowledges this concern, but this does not amend the fact that experimental validation is yet to include testing of predicted patient-specific drug sensitivity responses in ovarian cancer organoid model.

Our response: To address the concerns raised and further strengthen our findings in the ovarian cancer application, we have now expanded our analysis to include predictions for the most and least effective drugs in each of the three models by employing a similar approach to the one we used for the AML patient samples (Fig. 2b). It took us a considerable time to re-start growing the organoid lines, and make new treatment experiments in each line. We tested the predictions in the patient-derived PAX8+ cells to avoid the challenges of the fibroblasts, as pointed out by the reviewer, and which were acknowledged in the discussion.

We conducted a statistical comparison of the ex vivo drug sensitivity differences between the single-agent treatments predicted by the scTherapy model to be either effective or ineffective in the patient-derived PAX8+ cells. These comparisons yielded statistically significant differences ($p < 0.001$, Wilcoxon test), reinforcing the reliability of our predictions and the impressive utility of the model in distinguishing between effective and ineffective treatments in individual patients. These results are now added to **Figure 3** panel h, please see below.

Fig. 3 | Experimental validation in ovarian cancer patient-derived tumor organoids. (a-c) UMAP projection of the scRNA-seq transcriptomic profiles for the three HGSC patient samples, using standard Seurat integration workflow, where cell types were identified with ScType (left panel). Expression of the PAX8 marker, effectively separating tumor cells from the other cell populations (right panels). **(d-f)** Barplots showing cell inhibition differences between the patient-derived organoid cancer cells (PAX8+, blue bars) and non-cancerous normal cells (PAX8-, grey bars) for the 18 predicted multi-targeting drugs (left panels). The predicted effective doses are indicated in parentheses (μM), and the dotted vertical lines indicate 50% inhibition. The error bars represent SEM, based on three replicates of organoid treatments and curve-fitting in PAX8- cells. For patients 2 and 3 both organoids and the stromal cell cultures were available at the cell numbers sufficient for single-drug sensitivity and selectivity testing, whereas for Patient 1 the PAX8+ tumor cells originated from the patient organoid and PAX8- normal cells were available from additional Patients 4 and 5 (Suppl. Table 3). Statistical comparison of the treatment responses between PAX8+ and PAX8- cells across three HGSC patient samples with Wilcoxon test (right panels). **(g)** Representative immunofluorescent image of treatment-naive tumor organoids from Patient 1 sample. **(h)** Statistical comparison of ex vivo drug sensitivity differences in patient-derived PAX8+ cells between the treatments predicted by scTherapy to be either effective (n = 18) or ineffective (n = 10) in the individual patients (p < 0.001, Wilcoxon test). Suppl. Fig. 7 shows the same data in a heatmap, summarizing drug responses across multiple doses using drug sensitivity scores (DSS), instead of the percentage inhibition at the predicted effective dose (as shown here).

Authors could, for example, simply perform prediction for the individual cancer samples and then test the top effective treatments only in the organoid lines to assess response variability among 3 donors when subjected to specific treatments. If they can show that their bioinformatics pipeline can identify effective treatments in individual lines, that could considerably improve the data and increase the potential impact of the publication to extend conclusions to solid tumor malignancies.

Our response: In response to the reviewer’s great suggestion, we have now conducted additional experimental validations. We selected the top-treatments predicted to be the most effective in the individual ovarian cancer models, as detailed in **Figure 3** panels d-f, and then assessed experimentally the responses to these treatments across the organoid lines derived from all three samples. We also included 10 drugs predicted to be non-effective across all the samples as negative controls to evaluate the specificity of the model predictions.

The outcomes of these new experiments are depicted in the below heatmap (and added as new **Supplemental Figure 7**). This analysis did not only confirm the consistency and specificity of the model-predicted drug sensitivities, but also revealed distinct patterns of response variability among the HGSC patients, indicating a relatively heterogeneous drug sensitivity and resistance patterns across the patient-derived organoid lines. This further motivates the need for patient-specific and cancer-selective treatment predictions.

Suppl. Fig. 7 | Heatmap of measured drug sensitivities for treatments predicted either as effective or ineffective in ovarian cancer patients. Blue annotation in the left-hand column indicates that the drug was predicted to be effective in one or multiple ovarian cancer patients. Red indicates those drugs predicted as non-effective in all the patient samples. *Drug treatments for which high efficacy class was predicted by the scTherapy model, showcasing the accuracy of our predictive model across the three patient-derived organoid lines.

Minor point: The scalebar is missing in Figure 3e. These are new pictures presumably to avoid discussion regarding PAX8 negative cells, but now magnification is smaller, thus pictures are somewhat less informative regarding organoid morphology.

Our response: Thank you for pointing out the absence of a scale bar in the revised figures. We have now added the scale bar to Figure 3g to indicate a magnification of 250 μm .

Reviewer #3 (Remarks to the Author):

The revised version of the manuscript addresses most of my comments, and overall, I believe it has improved a lot in terms of readability and focus. I also appreciated the authors' effort to perform new analyses to extend the scope of the method beyond hematological malignancies.

Our response: We thank the reviewer for the constructive feedback and for the recognition of the efforts we have made to improve our manuscript and address the comments.

My remaining concern is related to the limited quantitative evaluation of the performance of the authors' method and its quantitative comparison and improved significance with respect to other state-of-the-art approaches in the field.

As stated in my original review, I understand that this is not straightforward, and I appreciate the new paragraph included by the authors in the discussion section where they conduct a review of the literature and mention possible competing methods.

I do not dispute the performance of scTherapy. However, I believe demonstrating the tool's performance is important to establishing its significance.

Our response: We appreciate the reviewer's continuous emphasis on a clear and more quantitative comparison to demonstrate the added value and significance of scTherapy. We have now conducted additional analyses and updated Figure 5 to provide a standard, more detailed comparative visualisation and quantification of the performance of the methods.

In the newly added panels c and d of **Figure 5**, we improved the quantitative evaluation by adding Receiver Operating Characteristic (ROC) curves and calculating Area Under the ROC Curves (AUC) values for each method. These standard performance metrics quantify the models' ability to discriminate between effective and ineffective drug treatments, using the experimentally measured drug sensitivity scores aggregated across the 12 AML patients. The AUC values (0.53 for BeyondCell, 0.57 for scDrug, and 0.71 for scTherapy) clearly demonstrate the superior predictive accuracy of scTherapy (**Figure 5d**). The improvement in

accuracy is statistically significant, as evidenced by DeLong test, where performance of scTherapy was significantly better compared to that of both scDrug and BeyondCell ($p < 0.01$).

We would like to point out further that unlike BeyondCell and scDrug, scTherapy does not only predict the efficacy of compounds, but it also determines the specific drug doses at which these compounds are likely to be effective. This dose-specific prediction capability is crucial for translational applications, where precise dosing is key to treatment outcomes. To ensure a fair comparison with the other methods, which do not offer dose differentiation, we focused on the overall drug responses of the top and bottom 15 drugs predicted as most effective and those predicted as least effective by each model for the individual patients. As part of the AML study programme, we have experimentally evaluated the efficacy of these drugs across multiple doses, as quantified by the drug sensitivity score (DSS, normalized area under the drug dose-response curve). Finally, we compared the DSS values across the models using pairwise Wilcoxon rank-sum tests, with p-values adjusted for the False Discovery Rate (FDR) using the Benjamini-Hochberg procedure (**Figure 5** panels a and b).

Fig. 5 | Quantitative comparison of monotherapy efficacy predictions in 12 AML patients. Drug sensitivity score (DSS)⁴⁵ distributions of the top-15 drugs predicted as (a) the most effective and (b) the least effective monotherapies by each model for individual patients. The predictions were compared using pairwise Wilcoxon rank-sum tests, with p-values adjusted for the False Discovery Rate (FDR) with the Benjamini-Hochberg procedure. *a model's predictions are significantly different ($p < 0.05$) compared to that of at least one of the two other methods; **a model's predictions show a significant difference ($p < 0.05$) when compared to both of the alternative methods. (c) Receiver Operating Characteristic (ROC) curves for each model demonstrating their ability to distinguish between effective and ineffective treatments based on the predictions of the most effective and least effective drugs by each model for individual patients (as shown in panels a and b). The shaded area around each curve represents the 95% confidence interval, illustrating the variability and reliability of the predictions based on data from 12 patients. (d) A summary table displays the Area Under the Curve (AUC) values for each model, quantifying their overall predictive accuracy. Statistical comparison was done with DeLong's test, indicating that the prediction performance

of scTherapy is significantly better than that of scDrug and BeyondCell ($p < 0.01$). The other two methods show statistically similar results ($p=0.29$).

We hope these enhancements in the quantitative and statistical evaluations now provide an easier comparison of scTherapy against the existing state-of-the-art methodologies in the field, and demonstrate its greatly added value in comparison to the other approaches.

I do not think this part should be mentioned only in the discussion, and I would ask the author to improve the visualization of Figure 5, as its current version does not allow for easy comparison among the methods.

Our response: We agree that the visibility and impact of our quantitative evaluations are crucial. Accordingly, we have moved the comparisons of scTherapy with existing methods from the discussion section into the main text (pages 11-12). Additionally, we have revised Figure 5 to include standard ROC curve comparisons, as well as statistical analyses, further illustrating the quantitative distinctions in performance between the methods (see above).

Reviewer #3 (Remarks on code availability):

I was only capable of performing a quick assessment, and the R repository seems ok.

Reviewer #4 (Remarks to the Author):

My critiques have been addressed

REVIEWERS' COMMENTS

Reviewer #2 (Remarks to the Author):

The authors have put major effort into addressing raised concerns regarding the applicability of their prediction pipeline on both AML and ovarian cancer. In particular new experimental data performed in the ovarian cancer organoid model (Figure 3h and Supplemental Figure 7), demonstrate clear differences between effective and ineffective treatments, confirming the validity of the bioinformatics approach and predictions. In its entirety, the study presents a novel and interesting tool that will certainly be of interest to the broad research community and could provide a positive impulse for advances in treatment options in oncology in general.

Reviewer #3 (Remarks to the Author):

All my remaining comments have been fully addressed.